# Misshapen coordinates protrusion restriction and actomyosin contractility during collective cell migration

Cédric Plutoni[1], Sarah Keil[1], Carlos Zeledon[1], Lara Elis Alberici Delsin[1], Barbara Decelle[1], Philippe P. Roux [1,2], Sébastien Carréno [1,2] & Gregory Emery [1,2]

Collective cell migration is involved in development, wound healing and metastasis. In the *Drosophila* ovary, border cells (BC) form a small cluster that migrates collectively through the egg chamber. To achieve directed motility, the BC cluster coordinates the formation of protrusions in its leader cell and contractility at the rear. Restricting protrusions to leader cells requires the actin and plasma membrane linker Moesin. Herein, we show that the Ste20-like kinase Misshapen phosphorylates Moesin in vitro and in BC. Depletion of Misshapen disrupts protrusion restriction, thereby allowing other cells within the cluster to protrude. In addition, we show that Misshapen is critical to generate contractile forces both at the rear of the cluster and at the base of protrusions. Together, our results indicate that Misshapen is a key regulator of BC migration as it coordinates two independent pathways that restrict protrusion formation to the leader cells and induces contractile forces.

---

[1] Institute for Research in Immunology and Cancer (IRIC), Université de Montréal, Montréal, QC, Canada. [2] Department of Pathology and Cell Biology, Faculty of Medicine, Université de Montréal, Montréal, QC, Canada. Correspondence and requests for materials should be addressed to G.E. (email: gregory.emery@umontreal.ca)

Collective cell migration plays crucial roles during morphogenesis, wound healing, and metastasis[1–4]. Border cell (BC) migration in the *Drosophila* egg chamber has emerged as a powerful model to study the collective migration of small cluster of tightly attached cells. BCs are somatic cells that detach from the follicular epithelium, form a small cluster of 6–10 cells and migrate between the giant nurse cells. Their migration is guided towards the oocyte, as it secretes ligands that activate receptor tyrosine kinases (RTK) on the cluster. These ligands target PVR (the sole PDGF-receptor and VEGF-receptor in *Drosophila*) and the EGF-receptor, which in turn activate the small GTPase Rac, thereby leading to the formation of actin-based protrusions[5]. The activation of the Myosin II regulatory light chain by the Rho-kinase Rok induces actomyosin contractility, which generates the traction and propulsive forces required for promoting BC motility[6,7].

Single cell migration studies have shown that a well-tuned coordination between protrusion dynamics at the front and contractility at the back is necessary for efficient cell migration[8]. A similar coordination may also control BC migration, as Myosin II activity is concentrated at the back and at the front of the cluster, while protrusion formation is largely limited to a single front leader cell. This coordination requires communication between the cells of the cluster[6,9]. Moreover, we previously found that the small GTPase Rab11, which is part of the endosomal recycling machinery, and the actin-binding and membrane-binding protein Moesin are required for this cell–cell communication[10]. In *Rab11* or *Moesin* loss-of-function conditions, multiple cells of the cluster form protrusions due to deregulated Rac activity. This coordination defect causes stalled migration due to opposing pulling forces. While the Rac activity restricting mechanism requires Rab11 and Moesin activity, the exact molecular pathway that achieves this restriction is unclear. The distribution of active Moesin at the periphery of the cluster suggests that it organizes a supracellular actin structure that unifies the cortices of the BCs. As Moesin increases cortical stiffness[11], it is possible that its cortical activation prevents protrusion formation by increasing cortical stiffness throughout the cluster periphery[12].

Since Moesin is crucial for the coordination of BC migration, we sought to understand its regulation. Activation of Moesin requires the phosphorylation of a conserved Thr residue within its actin-binding C-terminal ERM Association Domain (CERMAD, Thr556 in *Drosophila*)[13]. Here, we identifies Misshapen (Msn), a Ser/Thr kinase, as the direct activator of Moesin in BCs. While Msn was previously shown to be involved in BC migration, its mechanism of action was not defined[14]. We found that, through Moesin, Msn regulates protrusion restriction and cortical stiffness. Unexpectedly, we demonstrates that Msn is required for normal actomyosin contractility in a Moesin-independent manner. Overall, our findings show that Msn coordinates both protrusion formation and cortical contractility to promote the collective cell migration of BCs.

## Results

**Msn is a Moesin kinase required for BC migration.** We have previously shown that the mechanism that restricts Rac activation and protrusion formation to the leader cell of the BC cluster requires Moesin phosphorylation on Thr556[10]. To identify the Moesin kinase operating in BCs, we performed a targeted RNAi screen and scored candidates for migration defects (Fig. 1a and Supplementary Fig. 1a). Since most of the known kinases that activate the mammalian orthologs of Moesin (ERM proteins; Ezrin, Radixin, Moesin) belong to the Ste20 family, we used the *c306*-Gal4 driver to specifically deplete Ste20-like kinases in BCs. Of the 11 kinases tested, Msn and Tao depletion completely blocked BC migration while Pak3 and Hpo depletion partially affected BC migration. Surprisingly, depletion of Slik, the only Moesin kinase presently identified in flies[15,16], had no effect on BC migration (Fig. 1a).

To determine if one of these kinases activates Moesin in BCs, we measured the level of total Moesin and phospho-Moesin (pMoe) by immunofluorescence upon the depletion of candidate kinases. We found that only one of the two *msn*-RNAi constructs (RNAi #1) marginally reduced the total level of Moesin, while the other RNAis had no effect (Supplementary Fig. 1b). However, both RNAis against *msn* induced a strong reduction of pMoe levels at the cluster periphery (Supplementary Fig. 1b, c). While depletion of Tao resulted in a minor decrease of pMoe staining, depletion of Pak3 and Slik did not significantly affect pMoe levels (Fig. 1c). Overall, this demonstrates that Msn is essential for the normal phosphorylation of Moesin in BCs.

Next, we determined if Msn could directly phosphorylate Moesin. For this, we incubated immunoprecipitated Msn-HA from *Drosophila* S2 cells with the Moesin CERMAD domain produced in bacteria. While wild-type Msn was found to phosphorylate the CERMAD domain of Moesin, two different kinase-inactive Msn proteins[17,18] showed no activity towards Moesin (Fig. 1d and Supplementary Fig. 6), indicating that Msn directly regulates Moesin. More specifically, Msn directly phosphorylates the T556 residue of the CERMAD domain of Moesin, since a CERMAD where the Thr is mutated to an Ala is not phosphorylated by Msn in vitro (Supplementary Figs. 2a and 6).

To determine if the catalytic activity of Msn is required for BC migration, we performed rescue experiments in an Msn-depleted background. Using an RNAi-insensitive form of Msn, we found that expression of wild-type Msn (HA-msn) rescues BC migration, confirming that BC migration defects are due to Msn depletion. Next, we expressed an RNAi-insensitive and catalytically inactive mutant form of Msn (HA-msn[D160N]). To control that both HA-Msn and HA-Msn[D160N] are expressed at the same level, we checked their expression by Western Blot (Supplementary Figs. 2c and 7). We found that HA-Msn[D160N] was unable to rescue the phenotype associated with Msn depletion, demonstrating that the kinase activity of Msn is required for the migration of BCs (Fig. 1e). Using the same strategy, we found that the kinase activity of Msn is necessary for Moesin phosphorylation in vivo (Supplementary Fig. 2b).

We previously found that Moesin phosphorylation is restricted to the periphery of the BC cluster (Fig. 1f)[10]. To determine if Msn co-localizes with its substrate at this periphery, we used an enhancer-trap line that expresses Msn::YFP under the control of the endogenous *msn* promoter[19]. We found that Msn and Moesin co-localized in specific regions of the peripheral cortex of the cluster (Fig. 1f, arrows). Altogether, these data show that Msn phosphorylates Moesin to promote BC migration.

We previously showed that the small GTPase Rab11 is necessary for Moesin phosphorylation at the periphery of the cluster[10]. In addition to localizing at the cortex of the cluster, Msn is present on cytoplasmic punctae (Fig. 1g, i), suggesting that it is actively transported through vesicular trafficking. To test if Rab11 regulates the localization of Msn, we probed Msn localization in BCs expressing a dominant negative form of Rab11 (Rab11[S25N]). In wild type BCs, Msn accumulates at the cluster periphery (BC/NC (nurse cell) interface, Fig. 1g–e). Upon expression of Rab11[S25N], Msn is redistributed to the BC/BC interface (Fig. 1g–e), and its vesicular distribution is reduced (Fig. 1i). Rab11 interacts with the exocyst subunit Sec15 to form vesicles destined to the plasma membrane[20–22]. Interestingly, we found that Msn co-localizes with both Rab11 and Sec15 on vesicular structures (Supplementary Fig. 3a, b). These results suggest that Rab11 promotes the phosphorylation of Moesin[10] by regulating the vesicular transport of Msn to the cluster periphery.

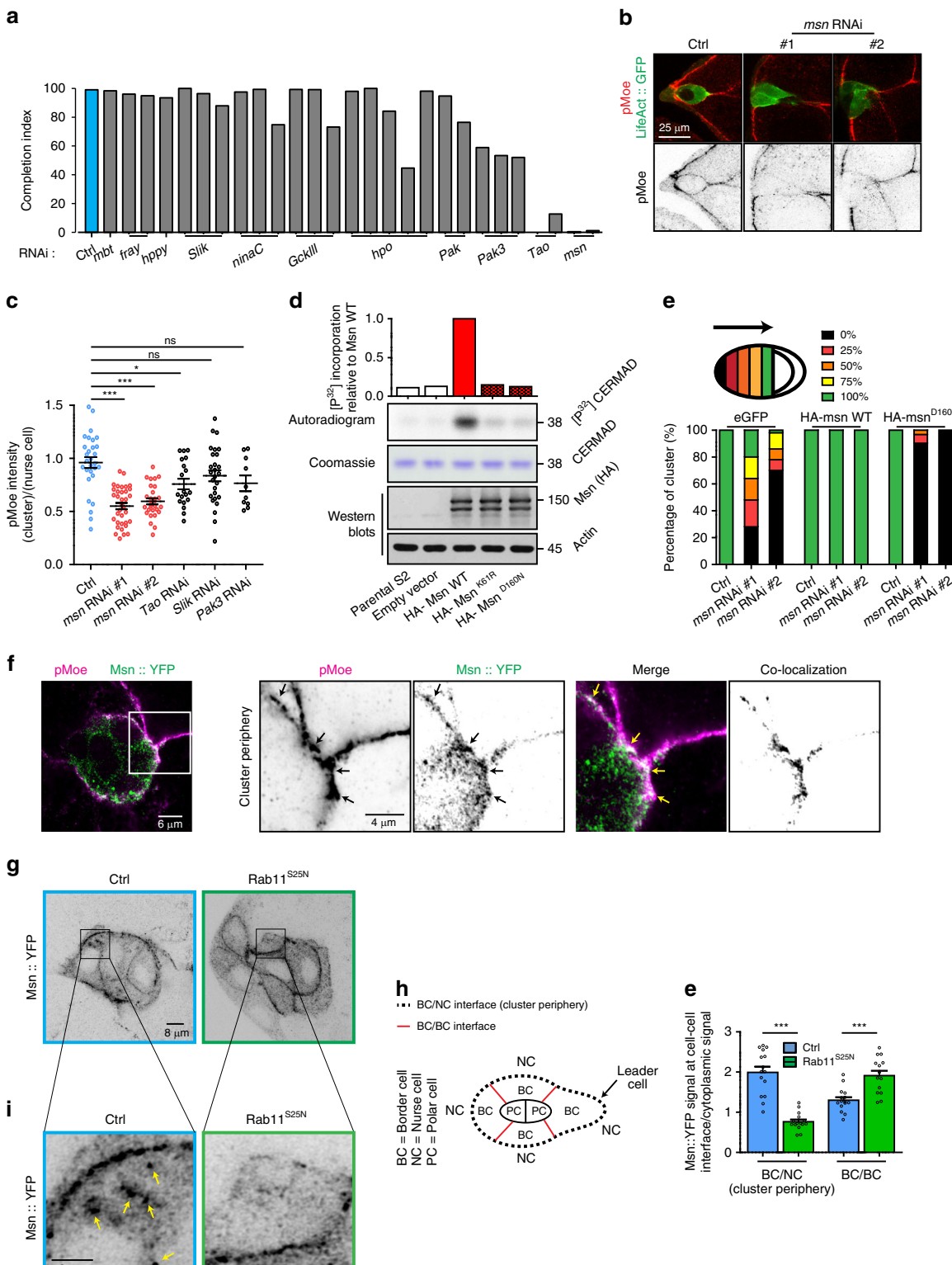

**Msn is required for BC protrusion restriction and detachment**. To characterize the role of Msn in BC migration, we analyzed time-lapse microscopy recordings of egg chambers incubated at 25 °C. At this temperature, which we used for every subsequent experiment, the depletion of Msn is partial, allowing for the observation of the various processes affected by Msn-depletion. Live imaging of BCs expressing LifeAct::GFP probe, shows that control clusters detach from the follicular epithelium, are motile and usually form a single prominent protrusion in the front cell (Fig. 2a–f). BCs depleted for Msn are able to form distinct clusters, but are less motile overall (Fig. 2a–c and Supplementary Movies 1 and 2).Importantly, they extend several prominent protrusions (Fig. 2d, e and Supplementary Movie 2) but remain attached to the follicle cells (Fig. 2d, f and Supplementary Movie 2). Based on these observations, Msn appears to be involved in the final step of BC detachment from the follicular epithelium and in the restriction of protrusions to the leader cell.

**Fig. 1** Misshapen is a Moesin kinase required for collective cell migration. **a** Migration indices determined after the depletion of the different Ste20-like kinases. Candidates were depleted using the RNAi lines indicated in the Supplementary Fig. 1, driven by *c306*-Gal4. This driver is employed for every subsequent experiment, except when specified. See Supplementary Fig. 1a for additional quantifications and details. **b** Representative images showing the intensity and distribution of pMoe at the onset of migration in control and Msn-depleted clusters. The pMoe channel is displayed individually as inverted greyscale images. **c** Quantification of the ratio of pMoe mean fluorescence intensity at the cluster periphery, normalized to the signal between nurse cells, after expression of mCherry RNAi ($n = 29$), *msn* RNAi#1 ($n = 35$), *msn* RNAi#2 ($n = 28$), *Tao* RNAi ($n = 19$), *Slik* RNAi ($n = 27$), and *Pak3* RNAi ($n = 10$). *n* represents the number of independent BC clusters. Non-significant (ns) $p > 0,05$, *$p < 0.05$; ***$p < 0.001$ (one way ANOVA test coupled with Bonferroni correction). Error bars show s.e.m. **d** Immunoprecipitated wild type and kinase-dead HA-tagged Msn were used in kinase reactions on the CERMAD domain of Moesin. Reactions were analyzed by autoradiography, Western blots, and Coomassie. **e** Quantification of the distance reached by BCs at stage 10 (0%, 25%, 50%, 75%, or 100% of the total distance) after the expression of a control RNAi or two independent RNAi against *msn*, together with eGFP, an RNAi-insensitive form of *msn* or a kinase-dead RNAi-insensitive form of *msn*, as indicated. $n = 143, 25, 50, 93, 42, 50, 35, 31$, and $17$, respectively, to the histogram order. *n* represents the number of independent egg chambers analyzed for the quantification. **f** Representative images showing the localization of Msn and pMoe in BCs. Their co-localization is highlighted by black arrows in separated channels (shown as inverted greyscale images) and yellow arrows in merged images. Co-localization images were obtained by superimposing the black and white negative images of Msn::YFP and pMoe signals.
**g** Representative images showing the localization of Msn in control clusters or after expression of a dominant negative form of Rab11 (*Rab11$^{S25N}$*), displayed as inverted greyscale images. **h** Schematic representation of border cell cluster with labeling of the different cells and cell interfaces. **e** Quantification of the ratio between the mean Msn::YFP fluorescence signal at the BCs interface within the cluster (BC/BC) or at the periphery of the cluster (BC/NC interface) ($n = 15$ independent BC cluster, for both condition. ***$p < 0.001$ (unpaired Student's *t*-test). Error bars show s.e.m. **i** Cropped regions from panel **c** showing the localization of Msn on vesicles (yellow arrows) in control BCs and *Rab11$^{S25N}$* expressing clusters,

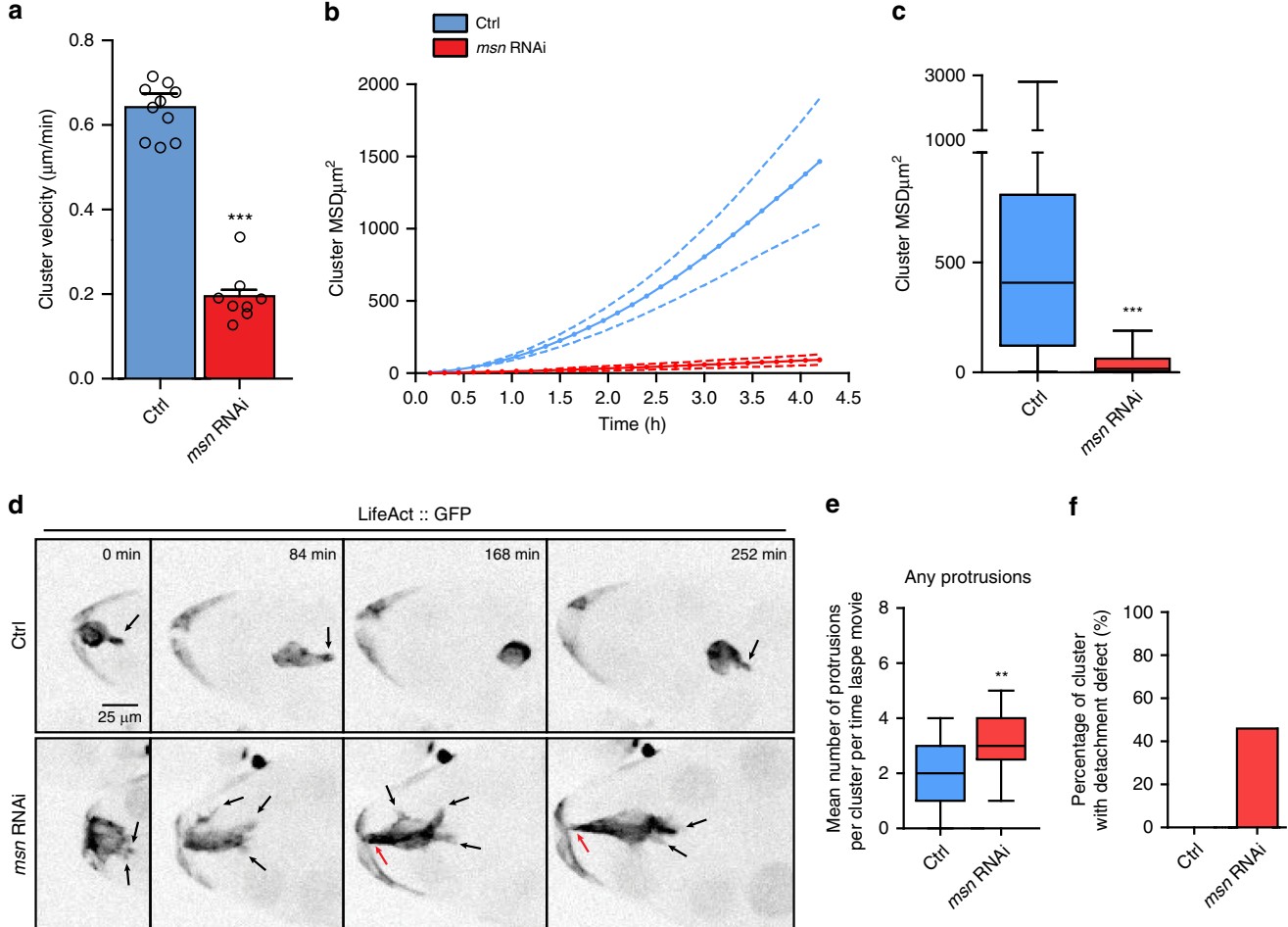

**Fig. 2** Msn is required for BC migration, protrusion restriction and detachment. **a** Quantification of cluster velocity of control ($n = 10$) and Msn-depleted clusters ($n = 8$) from live imaging. ***$p < 0.001$, unpaired Student's *t*-test. The error bars show s.e.m. Quantification of the mean square displacement (MSD) over time in **b** and cumulative in **c** of control ($n = 10$) and Msn-depleted clusters ($n = 8$) from live imaging. ***$p < 0.001$, two ways ANOVA test. On box plots, the box limits represent the first (lower) and the third (upper) quartile values, the whiskers show the smallest and the greatest values; and the center line represents the median. *n* represents the numbers of independent BC clusters. **d** Inverted still greyscale images from a time-lapse recording of clusters expressing *LifeAct::GFP* in control or Msn-depleted conditions. Black arrows indicate protrusions and red arrows point to the detachment defect (see Movie 2). **e** Using 2D live imaging, we quantify the number of protrusions per cluster at each time point of 5 h movies of control and Msn-depleted clusters ($n = 8$ and 10, respectively). Any protrusions larger than 4 μm were automatically detected using ADAPT plugin and LifeAct::GFP as a marker. **$p < 0.01$, two ways ANOVA test, The error bars show s.e.m. **f** Quantification of the percentage of clusters that started to migrate and still have not detached by stage 10 (fixed samples) ($n = 47$ and 39, for Control and *msn* RNAi respectively)

To determine how Msn restricts protrusion formation to the leader cell and promotes BC detachment, we analyzed the impact of its depletion on different parameters associated with protrusion formation and actomyosin contractility. To do this, we first performed detailed protrusion analyses. We generated 3D reconstructions and z-projections of both control and Msn-depleted clusters expressing the actin probe LifeAct::GFP (Fig. 3a and Supplementary Movies 3 and 4). This analysis revealed that

control clusters form a single large forward protrusion, while also forming small protrusions in non-leader cells (side protrusions, Fig. 3b). Msn-depletion lead to an increase in the total number of detected protrusions per cluster (Fig. 3b) but also to a change in protrusion morphology. Importantly, in control clusters, leader cell protrusions measure approximately three times the length of side protrusions. Msn-depletion promotes a reduction of this length ratio (Fig. 3c and Supplementary Movies 2–4). Hence,

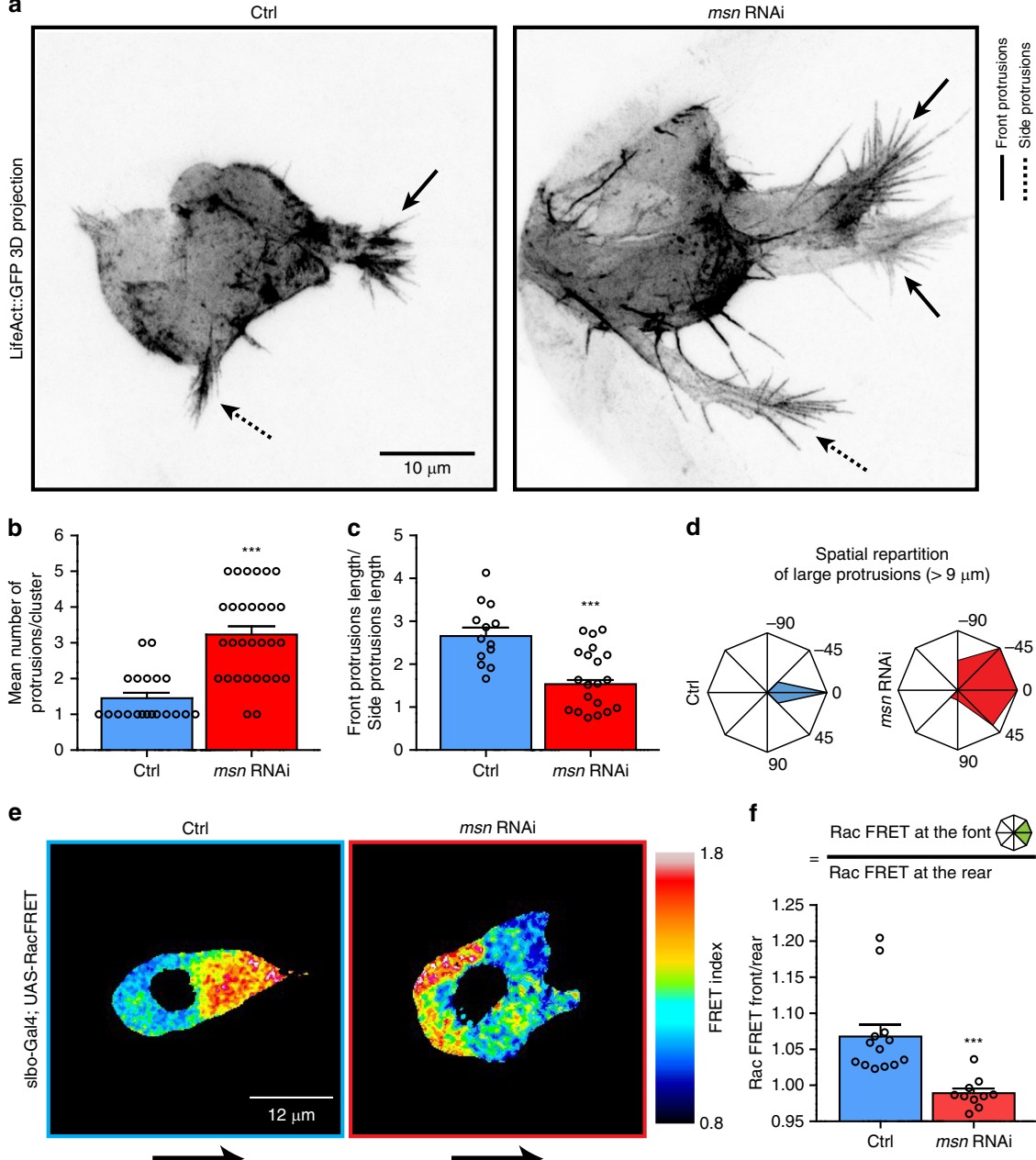

**Fig. 3** Msn restricts large protrusions and Rac activity to the leader cell. **a** Z-projection of *LifeAct::GFP* expressing control and Msn-depleted clusters shown in inverted greyscale. Plain arrows point to protrusions at the front of the clusters and dotted arrows point to protrusions at the side. 3D representation of the clusters are shown in Movies 3 and 4. **b** Quantification of the mean number of protrusions per cluster in both control and Msn-depleted clusters ($n = 20$ and $n = 30$, respectively). Any protrusions larger than 4 μm were manually counted using 3D fixed samples expressing LifeAct::GFP. **c** Ratio of front and side protrusion lengths in control and Msn-depleted clusters ($n = 13$ and 20, respectively). **d** Analysis of the orientation of protrusions longer than 9μm in control and Msn-depleted clusters ($n = 13$ and 20, respectively). **e** Processed FRET signal images from border cells expressing *UAS-Rac FRET* under the *Slbo*-Gal4 promoter in control and Msn-depleted clusters, at the onset of migration. FRET signal images are color-coded in 16 bit LUT from ImageJ. $n = 10$ for each conditions. Black arrows indicate the direction of migration. **f** Quantification of the ratio of the FRET indexes measured at front and back of control and Msn-depleted clusters ($n = 13$ and $n = 10$ independent BC cluster, respectively). Histogram represents the means of five different planes for each cluster. ***$p < 0.001$, unpaired Student's *t*-test. Error bars show s.e.m. $n$ represents the number of independent BC clusters

while in control clusters, long protrusions (>9 µm) are restricted to leader cells and point in the direction of the migration, long protrusion are formed by multiple BCs and point in different directions when Msn is depleted (Fig. 3d). These observations demonstrate that Msn restricts long protrusions to the leader cell.

Overall, our results suggest that Msn regulates the previously described Moesin-mediated cell–cell communication mechanism that controls protrusion restriction[10]. As this mechanism restricts high Rac activity to the front cell, we probed Rac activity in control and Msn-depleted clusters using a specific intra-molecular FRET biosensor[9,23]. We observed that Rac activity is restricted to leader cell in control clusters, leading to a polarization of the Rac activity in the direction of migration. We found that after Msn depletion, Rac activity is dysregulated and is no longer polarized (Fig. 3e, f).

These results show that Msn regulates cell–cell communication to restrict Rac activation and protrusion formation to the leader cell, possibly through the phosphorylation of Moesin.

**Msn regulates cortical stiffness across the BC cluster**. As Moesin increases cortical stiffness in its active state[11], we wanted to test if Msn depletion could alter cortical stability in BC clusters. Since membrane curvature is a direct hallmark of cortical stiffness and architecture[24], we performed a dynamic curvature analysis of migrating clusters (Fig. 4a). To allow visual analysis of curvature overtime, we constructed curvature heatmaps as kymographs (Fig. 4b). These kymographs reveal that the shape of the cortex of control clusters alternates between highly convex and highly concave curvatures (negative/positive/negative, NPN). This curvature NPN pattern indicates the presence of a protrusion. We found that NPN patterns are mostly restricted to the front of control clusters, while rare and transient NPN patterns are observed at their periphery (Fig. 4b, c). After depletion of Msn the number of NPN patterns increases. Furthermore, NPN patterns are more persistent and not restricted to the front of the cluster (Fig. 4b, c). This increase of the number of NPN patterns is consistent with reduced cortical stiffness. Interestingly, analysis of fixed Msn-depleted clusters reveals a decrease of sphericity compared to control clusters (Fig. 4d). As sphericity is another hallmark of stiffness[15], these results suggest that maintenance of cortical stiffness at the BC cluster periphery requires Msn.

Next, we investigated if Msn regulates the dynamics of cortical deformation by analyzing strong positive curvatures (>80°) on curvature maps. Automated detection of such curvatures revealed that control clusters have highly dynamic curvature deformation at the front of migration. In contrast, Msn depletion reduces the frequency of positive curvatures (Fig. 4e, f) and increases their lifetime (Fig. 4g), suggesting that Msn controls cortical deformation dynamics. Furthermore, this increased positive curvature lifetime corresponds to the static protrusions observed in time-lapse recordings of Msn-depleted clusters (Fig. 2d and Supplementary Movie 2).

Altogether, these results show that Msn regulates protrusion dynamics, in addition to overall cortical stiffness around the cluster.

**Msn regulates the spatial organization of actomyosin**. The detachment defect (Fig. 2d, f and Supplementary Movie 2) and the reduced dynamics of cortical deformation induced by Msn depletion (Fig. 4e, g) prompted us to determine if Msn might also regulate actin-mediated contractility. First, we used immunofluorescence to analyze the localization of the active, phosphorylated form of the regulatory light chain of Myosin-II (pMLC2). Previous work showed that pMLC2 is enriched at the rear of the cluster to promote detachment from the epithelium and at the front to promote protrusion contractility[6]. Our results show that Msn depletion does not alter the amount of pMLC2

present at the front or back of cluster (Fig. 5a, b), but affects the actomyosin organization in specific cortical domains (Fig. 5c–f).

Indeed, analysis of pMLC2 distribution revealed that it accumulates symmetrically at the base of the front protrusion and that this accumulation requires Msn (Fig. 5e, f). Furthermore, Myosin II is known to form transient *foci* at the cluster periphery (Fig. 5c, black arrows), which are necessary for cluster contraction[7,25]. We found that these *foci* are absent in Msn-depleted clusters (Fig. 5d). Finally, Msn-depleted clusters present an asymmetrical pMLC2 distribution along the axis of migration (Fig. 5d, red arrows). This abnormal actomyosin distribution may be at the root of cluster elongation by inefficiently spreading the contraction forces normally required for contractility[26].

Interestingly, we found that Msn localizes to cortical actin-rich structures (Supplementary Fig. 2a, b), including at the base and the tip of protrusions where it may regulate Myosin II activity (Supplementary Fig. 2a, c, d). This specific association with protrusions suggests that, in addition to potentially generating cortical stiffness via Moesin activation, Msn could also regulate protrusion retraction. Interestingly, a similar phenotype has been observed for Myosin II-depleted BC clusters[7] To test if the dysregulation of Myosin II affects the morphology of protrusions, we quantified the distance from the nuclei to the tip of the protrusion and found an elongation of both front and side protrusions after Msn depletion (Fig. 5g).

Overall, our data demonstrate that Msn is required for the spatial organization of actomyosin and for protrusion contractility.

**Msn coordinates contractility across the BC cluster**. The detachment defect and the actomyosin disorganization observed after Msn depletion (Fig. 2d, f and Supplementary Movie 2) prompted us to characterize the role of Msn in regulating contractility across the entire cluster. To assess the impact of Myosin-II dysregulation on contractility, we performed a morphodynamic analysis by mapping the extension and retraction dynamics of migrating clusters (Fig. 6a)[27]. Similar to the curvature analyses, we used color-coded kymographs to represent extensions and retractions (morphodynamic maps, MDMs) (Fig. 6b). These MDMs reveal that the front of control clusters protrude with brief moments of contraction, while retraction events occurs at the back, suggesting that extension and contraction events are coordinated across the cluster (Fig. 6b–e). Interestingly, after Msn depletion, extension, and retraction coordination is impaired. Indeed, Msn-depleted cluster MDMs reveal that at certain time points, the entire periphery of the cluster extend (Fig. 6b, red arrows), indicating that Msn is required to coordinate the cortical movements during BC migration. Quantification of extension and retraction velocities from the entire periphery of the cluster, shows that Msn depletion reduces these velocities as well as their mean variation (Fig. 6b–f), reflecting a global loss of the cluster periphery dynamics.

These observations are consistent with our previous results showing a loss of protrusion restriction after Msn depletion. It also strengthens our data showing that Msn controls cortical deformation. Overall, our findings suggest that Msn coordinates contractility in BC clusters through the spatial regulation of Myosin-II activity.

**Msn regulates protrusion restriction and BC contractility independently**. To determine if Msn regulates both protrusion restriction and actomyosin contractility through Moesin activation, we expressed a constitutively active form of Moesin ($Moe^{T556D}$) in Msn-depleted BCs. We found that the expression of $Moe^{T556D}$ rescued the Msn depletion-induced protrusion restriction defects, as shown by z-projections (Fig. 7a) and by the

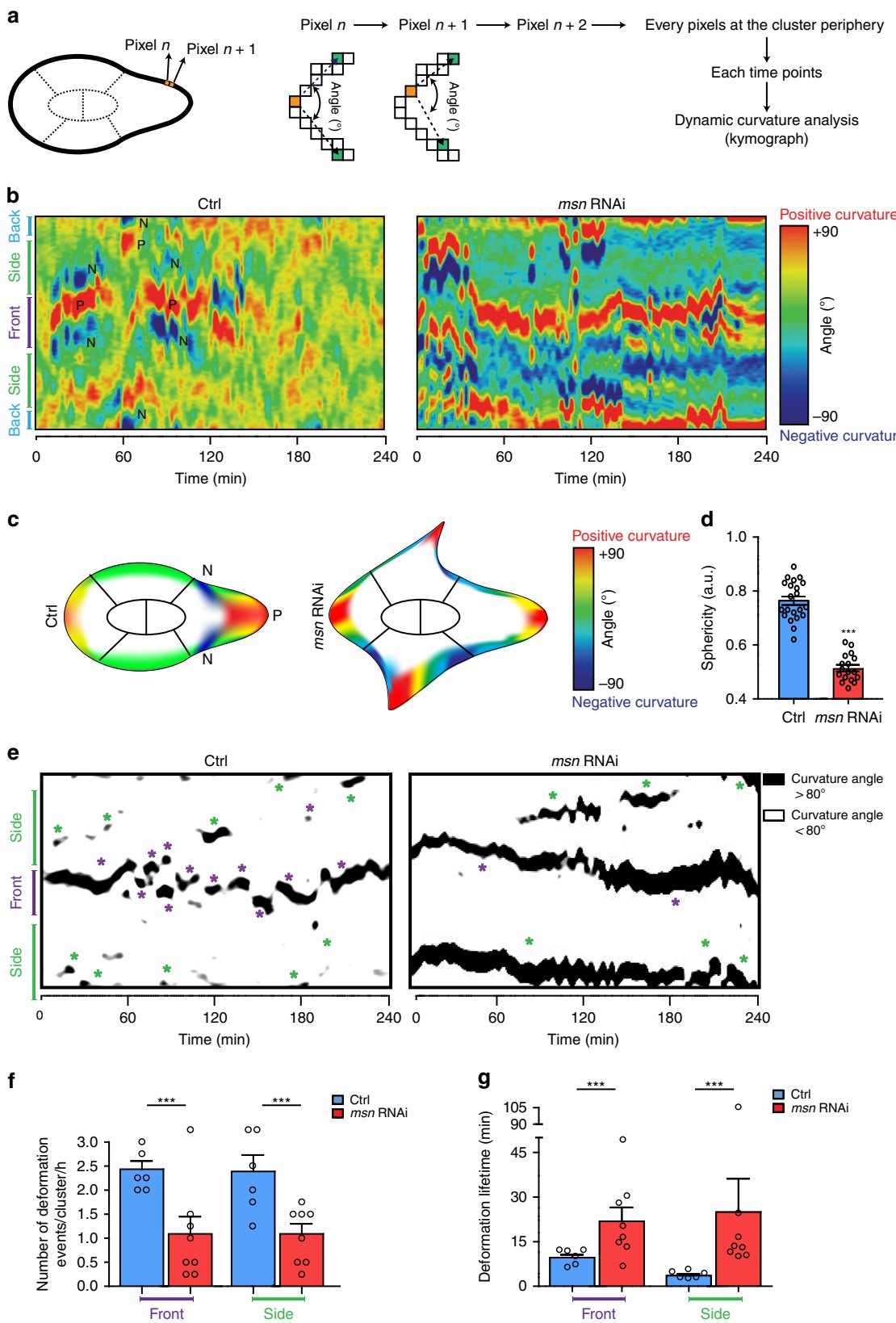

analysis of protrusions number and distribution (Fig. 7b, Supplementary Fig. 3a, b and Supplementary Movie 5). Furthermore, curvature analysis revealed that $Moe^{T556D}$ expression inhibits the ectopic deformations induced by Msn depletion (Supplementary Fig. 3c–e). These findings demonstrates that Msn regulates

protrusion restriction and cortical stiffness through the activation of Moesin.

However, $Moe^{T556D}$ expression rescued neither cluster detachment nor protrusion size, which are associated with contractility (Fig. 7a, c–e and Supplementary Movie 5). To verify that excessive

**Fig. 4** Msn regulates cortical stiffness. **a** Schematic representation explaining how the curvature analysis was performed by the ADAPT plugin of ImageJ. This analysis calculates the curvature at each pixel of the cell boundary as the angle ($\theta$) subtended by vectors (v1 and v2) to two points $n$ pixels away in each direction, for every time points. We represented the outcome as a heat map (positive curvature in red and negative curvature in blue) consisting of the kymograph of the different calculated curvatures. **b** Segmental curvatures calculated for each pixel at each time points are color-coded (red colors, positive curvature/convex; blue colors, negative curvatures/concave) and represented as a kymograph for control and Msn-depleted clusters. **c** Schematic representation of the curvatures observed in **b**. Extreme concave areas are restricted to the front of migration in the control clusters whereas they are found simultaneously at several positions of Msn-depleted clusters. **d** Cluster sphericity of control and Msn-depleted clusters ($n = 21$ and 17 independent BC clusters, respectively). **e** Representation of threshold curvature maps to highlight strong positive curvatures (>80º). Such curvatures are shown in black, while others are displayed in white. Events of strong positive curvature are automatically detected and indicated by purple asterisks at the front of the cluster and by green asterisks on the side of the cluster. **f** Quantification of the number of strong positive curvature events formed at the front and at the side of control and Msn-depleted clusters ($n = 6$ and 8 independent BC clusters, respectively). **g** Quantification of the strong positive curvature event lifetimes at the front and at the side of control and Msn-depleted clusters ($n = 6$ and eight independent BC clusters, respectively). $*p < 0.05$; $***p < 0.001$, unpaired Student's $t$-test. Error bars show s.e.m

activation of Moesin did not induce contractility defects per se, we expressed $Moe^{T556D}$ in a control background and observed no detachment defects (Supplementary Fig. 5c). To confirm that Moesin activation is not necessary for the Msn-dependent cluster contractility, we analyzed MDMs of clusters expressing $Moe^{T556D}$. These MDMs show that activity of Moesin is not sufficient to restore the contractility defect caused by Msn depletion (Fig. 7f–h). In addition, $Moe^{T556D}$ is unable to restore BC migration due to the strong Msn depletion-induced detachment defect (Fig. 7i–k). From these experiments, we concluded that active Moesin is sufficient to restrict protrusion formation and cortical deformation but is unable to promote contractility, suggesting that Msn acts through another pathway to regulate contractility.

As actomyosin contractility can be induced by the phosphorylation of MLC2 mediated by Rok (the Rock ortholog in *Drosophila*), we tested if increasing Myosin II activity is sufficient to restore contractility in Msn-depleted BCs. Thus, we expressed the catalytic domain of Rok ($Rok^{CAT}$) and found that it rescues the Msn depletion-induced global contractility defect (detachment, protrusion length, the contraction, and extension velocities defects) (Fig. 7a, c–h and Supplementary Movie 6). Interestingly, $Rok^{CAT}$ expression does not restore protrusion restriction (Fig. 7a, b and Supplementary Fig. 3a, b and Supplementary Movie 6), suggesting that this mechanism depends on Moesin activation but not on actomyosin contractility. To verify that excessive activation of Rok does not induce protrusion restriction defects per se, we expressed $Rok^{CAT}$ in a control background and observed no defaults (Supplementary Fig. 5c, d).

Curvature analysis revealed that expression of $Rok^{CAT}$ in Msn-depleted clusters restores the dynamic deformation of the cortex at the front and enhances dynamic deformations at the side of the clusters (Supplementary Fig. 4d, e). This suggests that Myosin II-mediated contractility controls the dynamics of cortex deformation, but is unable to restrict protrusion formation. This complementary analysis demonstrates that Msn controls protrusion restriction through Moesin activation and contractility through a Myosin II-dependent pathway (Fig. 8a, b).

## Discussion

The Ste20-like kinase MSn has been previously involved in different morphogenic events in *Drosophila*, including cell migration. Early work described the involvement of Msn in photoreceptor cell morphology and cytoskeletal dynamics[28]. Later, Msn was found to be involved in axon targeting and growth cone motility[29], nuclear migration in the eye[30], dorsal closure[17,29], elongation of the egg chamber through the migration of epithelial follicle cells[19,31], and in BC migration[14].

Here, we found that Msn acts on Moesin to restrict protrusion formation and regulates Myosin II to promote contractility

during BC migration. Previous work using *msn* loss-of-function mutants, found a complete block of BC migration attributed to an abnormal E-Cadherin distribution[14]. By using the UAS-Gal4 system to deplete Msn, we were able to analyze the function of Msn in BCs in greater details. Indeed, this methodology allows for selective RNAi expression in the BC cluster. It also allows for depletion modulation by incubating flies at various temperature. Similarly to the effect of *msn* mutants, severe depletion of Msn (29ºC) incurs a complete block of BC migration[14], while lesser depletions (25ºC) allowed for partial migration. On note, in the latter condition, we observed abnormally distributed E-Cadherin, but to the difference of *msn* mutant[14], we do not observe a reduction of the number of BCs per cluster. Nevertheless, our data suggests that it is not the prime cause of impaired migration, since distinct aspects of *msn* RNAi expression-induced migration defects are rescued via Moesin$^{T556D}$ or Rok$^{Cat}$ expression. Hence, while our findings do not exclude a downstream Msn activity on E-Cadherin, they strongly suggest that it is not its primary target.

Previous work in mammals showed that different kinases are able to phosphorylate ERM proteins[32]. In *Drosophila* however, Slik was so far the sole kinase known to act on Moe. Phosphorylation of Moesin by Slik occurs in the tip tracheal cells[33] and Slik was shown to be necessary for wing disc morphogenesis[15,16] by regulating epithelial integrity in vivo. Moreover, Slik-dependent phosphorylation of Moesin is necessary to regulate cortical stiffness during cytokinesis[11,34]. More recently, Slik has been described to act on Moesin to maintain polarity and cortical stiffness during asymmetric cell division[35].

Surprisingly, we found that Slik depletion has no impact on the phosphorylation of Moesin in BCs. This prompted us to perform a candidate RNAi screen to identify the Moesin kinase acting in BCs. As different kinases of the Ste20-like family act on ERMs in mammals[32], we screened among these for abnormal BC migration. Among the few kinases we identified as involved in BC migration, we found that Msn is required for Moesin phosphorylation in vivo and can phosphorylate Moesin in vitro.

Other Ste20-like kinases seem to play important functions during BC migration, including Pak3, Hpo, and Tao. While both the Hippo pathway[36–38] and Pak3[39] have already been shown to be involved in BC migration, the function of Tao is unknown.

Our previous work showed that Moesin is a key regulator of BC migration through its role in a cell–cell communication mechanism restricting protrusion formation to the leader cell[10]. As expected for a Moesin kinase, clusters depleted for Msn display large ectopic protrusions in follower cells due to the loss of cell–cell communication.

The mechanism by which Moesin and Msn inhibit the formation of large protrusions in follower cells is still unclear. Moesin regulates cortical stiffness, in accordance with a function of Msn in Moesin activation. When we analyzed the effect of Msn depletion, we found

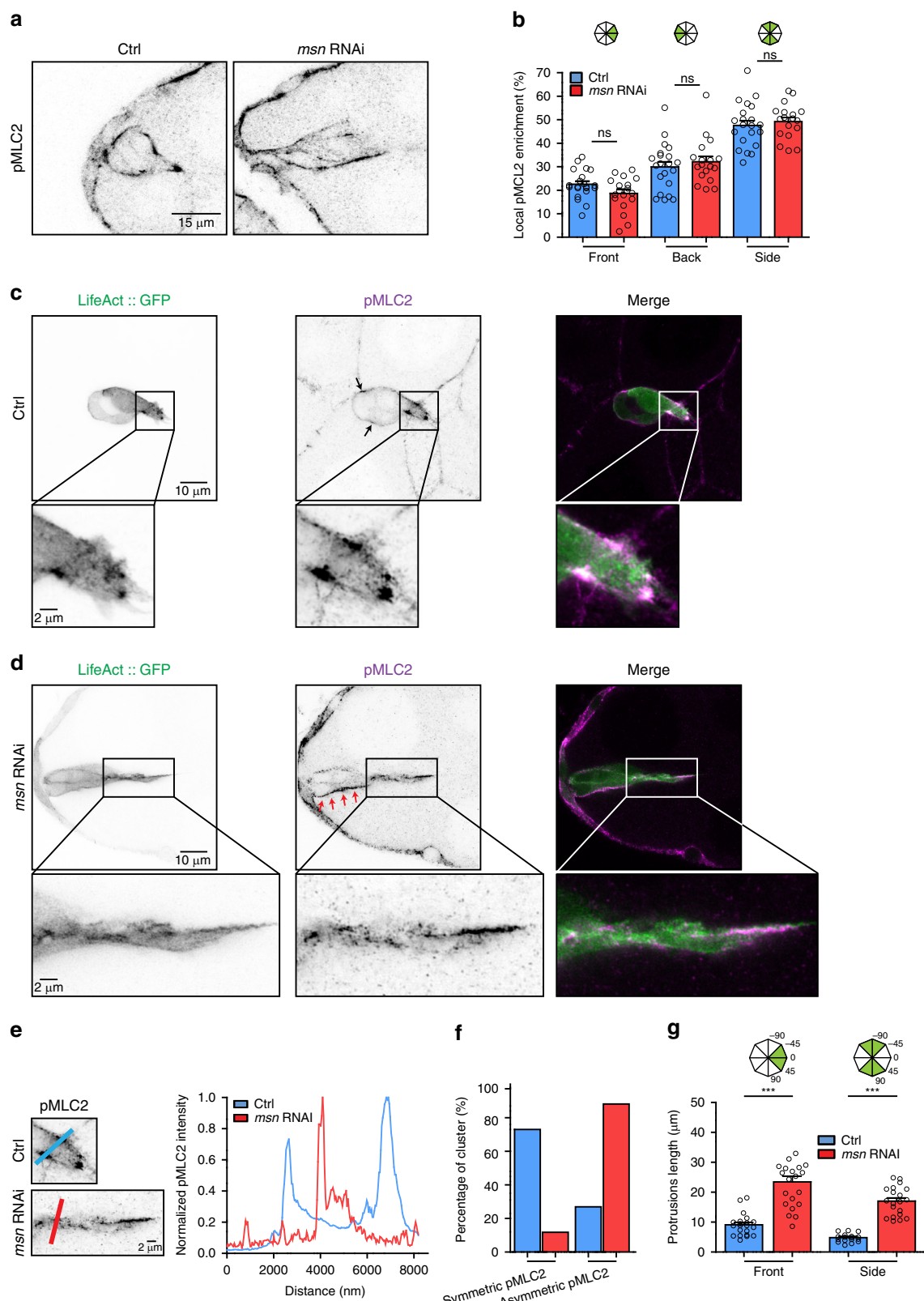

**Fig. 5** Msn regulates the spatial organization of the actomyosin cytoskeleton. **a** Inverted greyscale images of clusters stained with an anti-phosphorylated (Ser19) Myosin II Light Chain (pMLC2) antibody in control and Msn-depleted clusters. **b** Local pMLC2 enrichment is represented as the percentage of the signal at the front, back or side of the cluster over the signal of the entire cluster ($n = 21$ independent Control BC clusters and $n = 18$ independent *msn* RNAi BC clusters). **c** and **d** Representative images of control **c** and Msn-depleted clusters **d** expressing *LifeAct::GFP* and stained for pMLC2. Black arrows point to pMLC2 foci in the control cluster and red arrows to the continuous lateral accumulation of pMLC2 along the Msn-depleted cluster. **e** Analysis of the pMCL2 distribution in the front protrusion in control and Msn-depleted clusters. **f** Percentage of clusters with symmetric or asymmetric enrichment of pMLC2 at the protrusion basis ($n = 21$ independent Control BC clusters and $n = 16$ independent *msn* RNAi BC clusters). **g** Length of front and side protrusions in control and Msn-depleted clusters ($n = 20$ independent BC clusters, for both conditions). Non-significant (ns) $p > 0.05$, ***$p < 0.001$, unpaired Student's $t$-test Error bars show s.e.m

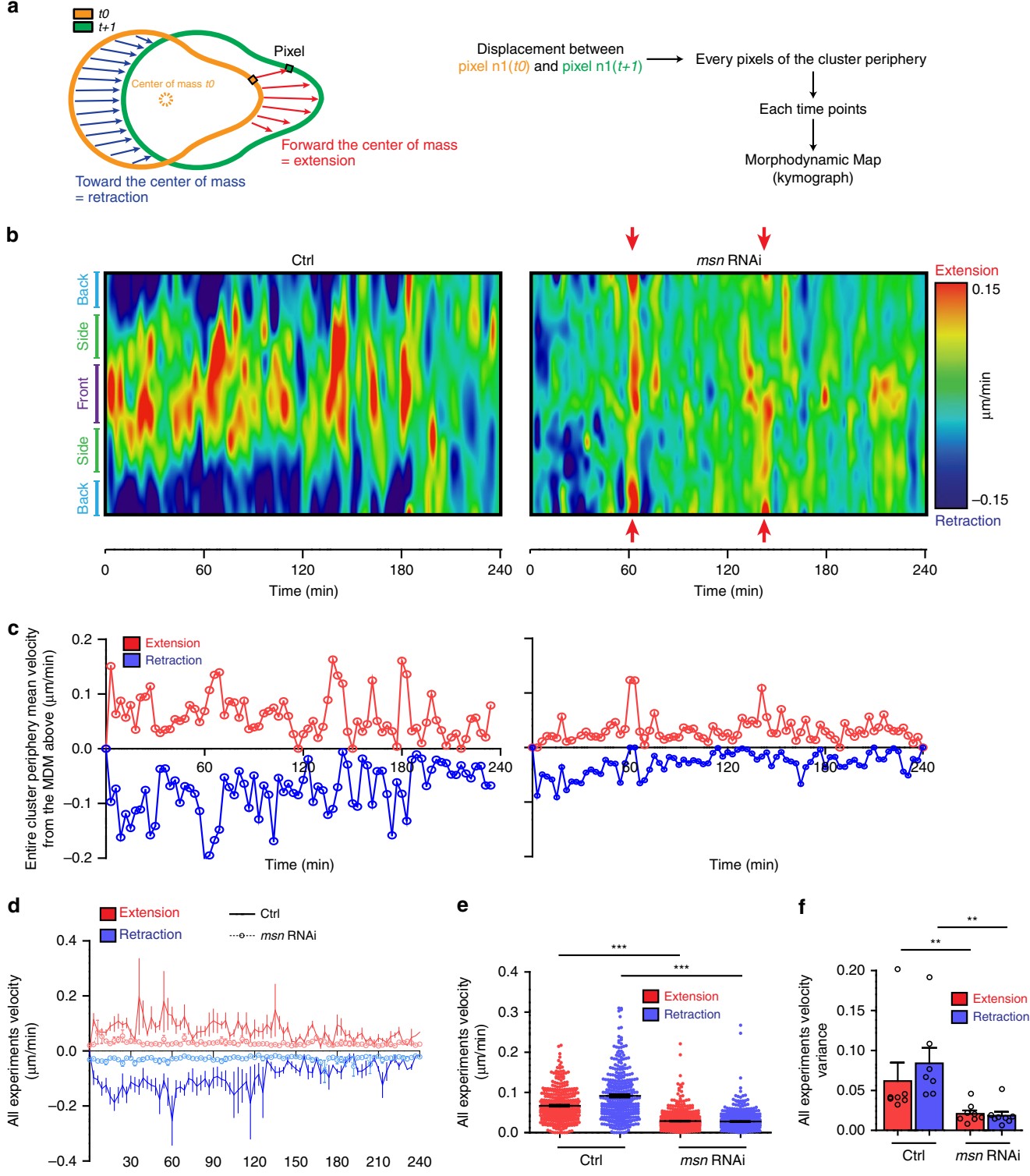

a reduction of sphericity and curvature, two morphological parameters associated with stiffness[24]. Prior work in MDCK cells showed that actin supracellular structures lead to cortical stiffness and inhibition of protrusion formation[40]. Hence, we are hypothesizing that Msn and Moesin inhibit protrusion formation by ensuring a continuous cortical stiffness across the different cells of the cluster.

Our previous work on Rab11 showed that it is involved both in the regulation of the polarization of the activity of guidance RTKs

PVR and EGFR during BC migration and in the phosphorylation of Moesin[10,41]. The effect on Moesin seemed epistatic, as the expression of an active form of Moesin was able to rescue the effect of dominant negative Rab11[10].

Interestingly, after Rab11-dominant negative expression, Msn relocalized from the cluster periphery to cellular interface inside the cluster. Surprisingly, despite being present on these interface, Moesin is not phosphorylated following Rab11-dominant

**Fig. 6** Msn is required for border cell contractility. **a** Schematic representation explaining how the morphodynamic analysis of migrating clusters was performed using the ADAPT plugin of ImageJ. The plugin measures the displacement of the overall cluster according to its geometric center (orange dotted circle) and represents these quantifications as a color-coded kymograph. On thes schemes, negative velocities/retraction events are represented by blue arrows whereas positive velocities/extensions events are represented by red arrows. **b** Morphodynamic map displays as a kymograph the segmental velocities calculated and color-coded for visualization, for each position of the periphery of a cluster (red colors, extension; blue colors, contraction). The center of the MDM depicts the dynamics at the front of the cluster (purple) while the side (light blue) and the back (green) of the cluster is split at both the top and bottom extremities of the MDM. **c** Velocities measured at each time point of the corresponding morphodynamic map shown in **b** (left, control cluster, right, Msn-depleted cluster). **d** Mean velocity values over time for extensions (red) and contractions (blue) in control (filled line, n = 7 independent BC clusters) and Msn-depleted clusters (dotted line with circles, n = 8 independent BC clusters). **e** Dot plot of each velocity values for each times point for each movies (for extensions and contractions) ***p < 0,001, two ways ANOVA test. Error bars show s.e.m. **f** Histogram representing the mean variance for extensions and contractions velocities. Unpaired Student's t-test, **p < 0.01. Error bars show s.e.m. (n = 7 independent Control clusters and n = 8 independent Msn-depleted clusters)

negative expression[10]. Moesin exists in a closed and an open conformation. Previous work demonstrated that ERM proteins bind to Phosphatidylinositol 4,5-bisphosphate (Ptd(4,5)P$_2$). Upon binding, ERMs partially change conformation and become accessible to phosphorylation by the kinase LOK, one of the Slik orthologs in mammals. Lok acts as a wedge to pry open and subsequently phosphorylate ERM proteins[42]. Msn lacks this wedge domain found in LOK and thus, may require a co-factor to efficiently phosphorylate Moesin at cellular interface inside the cluster. Alternatively, Ptd(4,5)P$_2$ might be enriched at the periphery of the cluster.

In addition to its direct action on Moesin, Msn is also required to induce actomyosin contractility. Msn depletion induces detachment defects, elongated protrusions and a general lack of apparent contractility. While expression of active Moesin in Msn-depleted clusters rescues parameters associated with protrusion restriction and cortical stiffness, it does not influence those associated with contractility. On the other hand, expression of the Rok catalytic domain in Msn-depleted clusters rescues overall contractility, while not impacting protrusion restriction to leader cells.

Future work is required to determine the mechanism by which Msn affects contractility, in particular to identify its target in this pathway. While we found that Msn is involved in the spatial organization of Myosin II activity, it seems unlikely that it is involved in its direct activation, as Msn depletion does not affect total pSqh levels. Observation made by others suggest that a precise distribution of the actomyosin cytoskeleton at the multicellular level is important for its processivity and efficient cell retraction[26]. Our observation that Msn depletion leads to a disorganized actomyosin cytoskeleton at both the base of protrusions and the periphery of the cluster suggests that Msn is necessary to ensure Myosin II processivity and global cell contractility.

To migrate, cells need to form extensions at the front and to contract at the back. These two events are coordinated both in individually migrating cells and in collectively migrating cell clusters. Few molecular mechanisms are known that coordinate these two processes, in particular during collective migration. Recent work demonstrated that a mechanism similar to planar cell polarity acts simultaneously to control the formation of protrusions and promote contractility in the follicle epithelium surrounding the *Drosophila* egg chamber[43,44]. However, during the migration of follicular epithelium cells there is no hierarchy between the different migratory cells. Rather, in BC clusters the important difference rests in the cell behaviors of leader cells producing dynamic protrusions and follower cells contracting their trailing edge. Our work demonstrates that a single kinase is able to control protrusion formation and global contractility simultaneously. We propose that the supracellular actin structure regulated by Msn and Moesin serves to coordinate extension and retraction events. In this process, E-cadherin-bases cell–cell junctions might help to connect the actin structure of individual cells and thereby helping the mechanical transfer of information, in accordance to recent findings by others[45].

The relevance of this observation to other collective migration systems, in particular in higher eukaryotes, is still to be determined. Interestingly though, recent studies have identified a function for Msn orthologs in various collective cell migrations. For example in Zebrafish, Msn1 was shown to act on actomyosin contractility during dorsal closure in vivo[46]. In mammals, the ortholog MAP4K4 regulates the endothelial sprouting and protrusion formation in angiogenesis assays and in vivo neovascularization by regulating the dynamics of focal adhesions through Moesin phosphorylation[47]. Together with our findings, these studies suggest that Msn might be a central regulator of various types of collective migrations.

## Methods

**Drosophila genetics**. To identify Ste20-like kinases involved in BC migration, the UAS-RNAi transgenic lines listed in Supplementary Fig. 1 were driven by *c306-Gal4;UAS-LifeAct::GFP* at 29 °C for 48 h. At least two independent RNAi lines (from the Vienna *Drosophila* RNAi Center and/or the Bloomington TRiP Stock Collection) were used for each gene. Other stocks used were *msn::YFP/TM6* (DGRC #115454, gift from Sally Horne-Badovinac, University of Chicago[19,48]), *UAS-Rab11$^{S25N}$* (gift from Marcos Gonzalez-Gaitan, University of Geneva[49]), *UAS-RacFRET/TM3* (bl#31431, gift from Denise J. Montell, UCSB), UAS-Sec15::mCherry (gift from Christian Bökel, TU Dresden[50]). The following stocks were acquired from the Bloomington Stock Collection *UAS-Moesin$^{T559D}$::Myc* (bl#8630), *UAS-Rok$^{CAT}$* (bl#6669), *UAS-Lifeact::GFP* (bl#35544), *UAS-LifeAct::Ruby* (bl#35545) and UAS-eGFP (bl#6658). The two *msn* RNAi lines used were *msn$^{JF03219}$* from the Bloomington TRiP collection (RNAi #1, bl#28791) and *msn$^{KK108948}$* from the Vienna Drosophila RNAi Center (RNAi #2, v101517). As control, we used an RNAi against mCherry (bl#35785). All the crosses with the *msn* RNAi (except for the Ste20-like kinases screen done at 29 °C) were done at 25 °C and maintained at that temperature until dissection. Transgenic flies overexpressing the RNAi-insensitive variants of *msn* were generated by Bestgene Inc., by injecting a pUAS-attB plasmid containing a modified HA-msn sequence (see below).

**Heat-shock experiments, mRNA extraction, and QPCR analysis**. Female were heat-shocked at 37 °C for 1 h and then incubated at 30 °C for 72 h and directly dissected for mRNA extraction using RNeasy Mini kit (Quiagen).

**Msn kinase dead mutants and RNAi insensitive mutant**. *pA-HA-msn vector*: Msn was subcloned from pOT2-msn (clone LD34191, gift from Vincent Archambault, IRIC, Montreal) into the pAHW gateway vector generating the pA-HA-msn vector. Those were used for in vitro kinase assays. First, we have created pDNR-msn using the following primers:

5′-GGGGACAAGTTTGTACAAAAAAGCAGGCTTCATGGCGCACCAGCAGCAACA-3′

5′-GGGGACCACTTTGTACAAGAAAGCTGGGTCTTACCAATTGGCCATGCCCG-3′

We have created the K61R and the D160N using QuickChange mutagenesis on the pDNR-msn plasmid and we have inserted it into the pAHW plasmid as below. To mutagenize the pDNR-msn sequence, we have used the following primers:

For the K51R mutation:

5′-GGTCAATTGGCTGCCATAcgcGTGATGGACGTCACC-3′

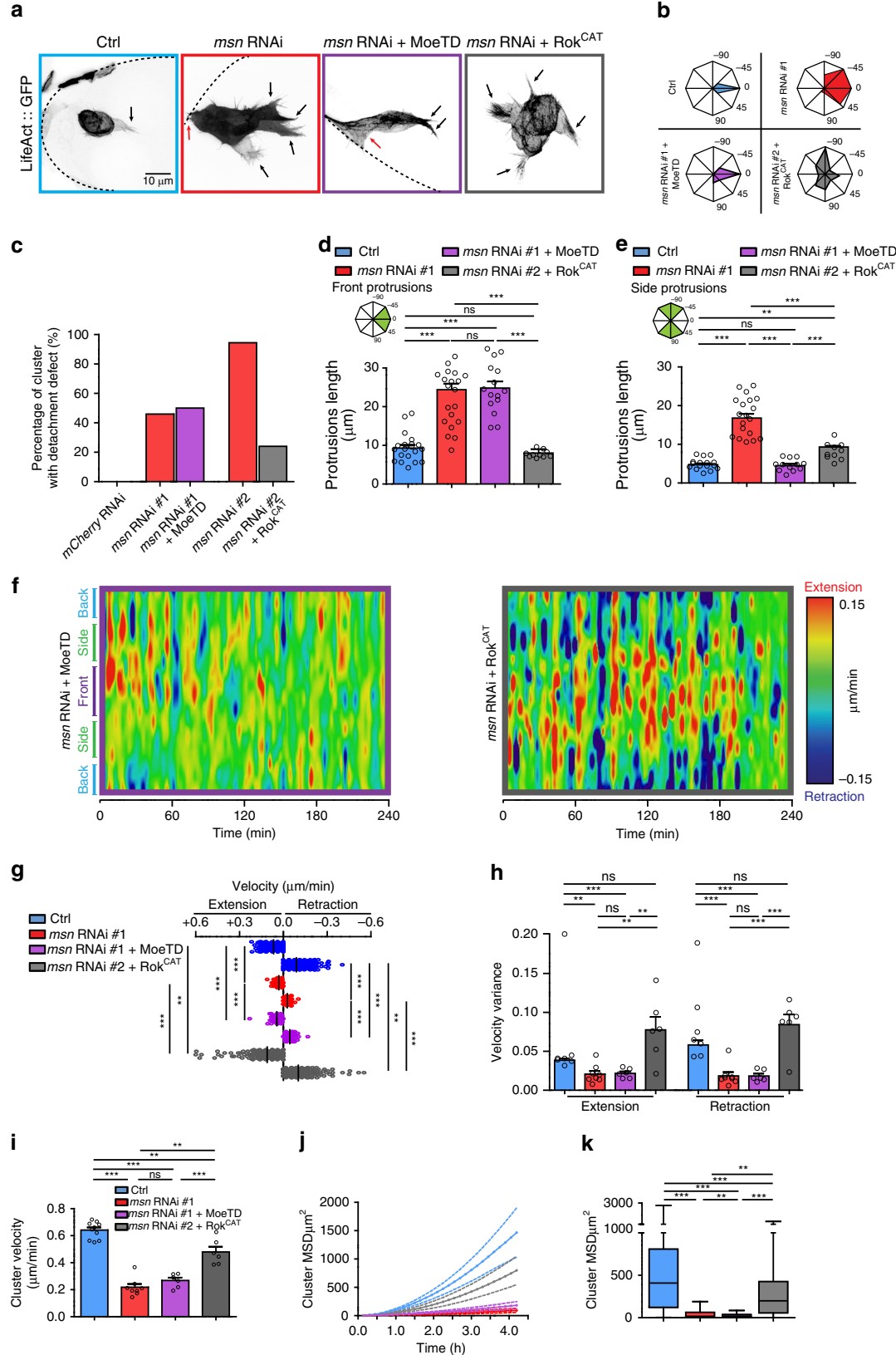

5′-GGTGACGTCCATCACgcgTATGGCAGCCAATTGACC-3′
For the D160N mutation:
5′-GTCATCCATCGCaacATCAAGGGGCAGAATG-3′
5′-CATTCTGCCCCTTGATgttGCGATGGATGAC-3′
*HA-msn-RNAi insensitive*: We have generated the HA-msn-RNAi insensitive (wild type and kinase dead mutants) from the pA-HA-msn plasmid. The RNAi-insensitive mutants contain silent, point mutations in the regions

targeted by both RNAi lines (from 2483 to 2945 bp for RNAi #1 and from 2279 to 2428 bp for RNAi #2). We have ordered the insensitive sequences to gBloc, Integrated DNA Technologies (IDT). Those mutated regions, were used to replace the corresponding *msn* region using standard cloning methods.

*UAS-HA-msn-RNAi insensitive*: We have subcloned HA-msn (wilt type and kinase dead mutants) from pA-HA-msn plasmid into the pUAS-AttB-Mek

**Fig. 7** Msn regulates protrusion restriction and contractility, independently. **a** Representative z-projections of clusters expressing *LifeAct::GFP* in control conditions, after depletion of Msn or in Msn-depleted clusters expressing *Moe*[T556D] or *Rok*[CAT]. Black arrows point to protrusions and red arrows to detachment defects. **b** Analysis of the orientation of protrusions longer than 9 µm in control conditions ($n = 13$), after depletion of Msn ($n = 20$) or in Msn-depleted clusters expressing *Moe*[T556D] ($n = 15$) or *Rok*[CAT] ($n = 10$). **c** Quantification of the percentage of clusters that started to migrate but have a detachment defects at stage 10 ($n = 47, 39, 124, 35,$ and 50 independent BC clusters, respectively to the histogram order). **d** and **e** Length of front and side protrusions were determined in in control conditions ($n = 20$), after depletion of Msn ($n = 20$) or in Msn-depleted clusters expressing *Moe*[T556D] ($n = 15$) or *Rok*[CAT] ($n = 10$). **f** Morphodynamic maps for each condition were built as in Fig. 4. **g** Dot plot of velocity means for extensions (red) and contractions (blue) at each time point measured ($n > 360$). **h** Histogram representing the variance of the overall velocity values for extensions (red) and contractions (blue) in control conditions ($n = 7$), after depletion of Msn ($n = 8$) or in Msn-depleted clusters expressing *Moe*[T556D] ($n = 6$) or *Rok*[CAT] ($n = 6$). **i–k** Quantification of cluster velocity **i** and of the mean square displacement (over time in **j** and cumulative in **k**) in control conditions ($n = 10$), after depletion of Msn ($n = 8$) or in Msn-depleted clusters expressing *Moe*[T556D] ($n = 6$) or *Rok*[CAT] ($n = 6$). Non-significant (ns) $p > 0.05$; **$p < 0.01$; ***$p < 0.001$, one way ANOVA test coupled with Bonferroni correction. On histograms, error bars show s.e.m. On box plots, the box limits represent the first (lower) and the third (upper) quartiles, the whiskers show the smallest and the greatest values; and the center line represents the median. $n$ represents the number of independent BC clusters

plasmids (from Marc Therrien's laboratory). Those constructs were used to generate UAS-HA-msn-RNAi insensitive flies for rescues experiments.

*Ste20 like kinase screen*: Ste20-like kinase were depleted as described in the "*Drosophila* genetics" section of the "Methods". Females were maintained at 29 °C and fed with yeast for 48 h prior to dissection. Egg chambers were dissected from adult young female in PBS 1X and fixed 20 min in 4% paraformaldehyde 4% diluted in PBS. Egg chamber were stained and mounted using standard immunofluorescence protocol[41]. BC migration was quantified at stage 10 of egg chamber development using an inverted microscope (Leica DM IRB). The Migration Index (M.I.) was calculated with the following formula:

$$\text{M.I.} = \frac{(1 * n(100\%) + 0.75 * n(75\%) + 0.5 * n(50\%) + 0.25 * n(25\%) + 0 * n(0\%) + 0.5 * n(\text{split} - \text{clusters})}{n(\text{total})}$$

$n(100\%)$ corresponds to the number of egg chambers where the cluster reached the oocyte, $n(75\%)$, the number of chambers where the cluster migrated to 75% of the final distance, etc. We also considered in our quantification, clusters that split and where three or more cells did not migrate whereas the rest of the cluster reached to the oocyte. We gave a 0.5 coefficient to these clusters in our migration index, since approximately half the cells only completed migration. A M.I. of 1 means 100% of the cluster has completed migration, whereas a value of 0 means that none of the clusters in the examined egg chambers migrated. A value of 0.5 could be obtained by various combinations of migration defects, including all the egg chambers displaying 50% migration or half of the chambers displaying a 100% migration and half of the chambers where the clusters did not migrate. The Completion Index (C. I.) corresponds to the number of egg chambers where the migration was completed ($n(100\%)$) divided by the total number of egg chambers ($n(\text{total})$):

$$\text{C.I} = \frac{n(100\%)}{n(\text{total})}$$

**Live Imaging and Rac FRET analysis**. Living egg chambers were prepared for real-time imaging using the protocol developed by Denise Montell laboratory[9] and imaged on a spinning-disk confocal (Zeiss) equipped with the camera AxioCam 506 Mono from Zeiss and a ×20/0.8 PlanApochromat objective. FRET imaging of live BCs was acquired on an Zeiss LSM880 inverted confocal microscope (Zeiss) equipped with a ×40/1.3 PlanApochromat oil immersion objective. Both CFP and YPF were captured simultaneously with independent PMT detectors. One image every 3 min were acquired. FRET images were generated from the CFP and YFP images using a homemade macro for ImageJ (gift from Damien Ramel, Université Paul Sabatier Toulouse III). Briefly, a Gaussian smooth filter was first applied to both channels. Then, the YFP image was used to create a binary mask with background set to zero. This mask is applied to the original smoothed CFP and YFP to remove the background signal. The final image ratio was generated by dividing the YFP signal by that of CFP with the «divide» function of ImageJ. The FRET ratio was calculated in the entire BC cluster or specifically at the leading edge and at the back of the cluster by measuring the average intensity of FRET and CFP using the Region measurements tool of ImageJ. For quantification, we used five different stacks and compare the means.

**Morphodynamics and curvature analysis**. Morphodynamics and curvature maps were generated using the open source ADAPT plugin, created and published by David J. Barry et al.[27]. The analyses were done on a single plane from confocal movies using a spinning-disk confocal (Zeiss) equipped with the camera AxioCam 506 Mono from Zeiss and a ×20/0.8 PlanApochromat objective. One image every 3 min were acquired. Three different planes were recorded. From these, we choose the plane with the best signal to noise ratio to perform the analysis. All velocity values are directly measured by the plugin and plotted in GraphPad Prism. For visual representation, raw map images (in.tif format) were stretched to optimize visualization. To detect membrane deformation events, we used the curvature map

from the same ADAPT plugin. We applied a threshold in order to render the images into a binary of two population of curvatures, strong positive (>80°) in black and the rest in white. The "particle analysis" tool from ImageJ was used to automatically assess the number of deformation events at the front and at the side of the cluster. For control conditions, time zero corresponds to the moment when the cluster visibly detached from the epithelium. As Msn-depleted BCs rarely detach, the time zero corresponds to the moment when the cluster is formed and rounded.

**Measurements of cluster velocity and mean square displacement**. Automated cell tracking clusters were performed by using the "Cell tracker" plugin in ImageJ. Velocities were calculated using Ibidi's Chemotaxis tool (http://ibidi.com/software/chemotaxis_and_migration_tool/). Mean square displacements were calculated using an homemade MSD plugin of Excel developed by R. Gorelik and A. Gautreau[51].

**Measurement of cluster sphericity, protrusion distribution, number, and length**. 3D reconstructions, representations, cluster sphericity, and protrusion length measurements were performed in Imaris (Bitplane) from 3D volume reconstructions of LifeAct::GFP and DAPI, both acquired with the Airyscan detector of a Zeiss LSM880 confocal. Sphericity was measured in the statistical analysis module of Imaris. Protrusion lengths of fixed samples were determined by measuring the distance between the center of the nucleus (DAPI) and the tip of the protrusions (LifeAct::GFP) for each protruding cell. Six measurements were performed at different positions of each protrusions and the mean was plotted in GraphPad PRISM. For protrusion number quantifications on fixed samples, a circle was drawn around the cell body. Any actin extension longer than 4 µm beyond this circle was defined as a protrusion. Extension longer than 9 µm was defined as "main protrusion". For distribution analysis, each protrusion defined as previously was aligned on a radar map divided into eight different sectors of 45° each with the leading edge set at zero degrees and the trailing edge at 180°. The length and the width of the arrow represent the amount of protrusions in a given direction. Analysis and quantification were done with ImageJ and representations with Excel. For protrusion number quantification on live samples, we quantify the number of protrusions per cluster at each time point of 5 h movies. Any protrusions larger than 4 µm were automatically detected using ADAPT plugin of ImageJ and *Life-Act*::GFP as a marker. Analysis and quantification were done with ImageJ and representations with GraphPad PRISM.

**Quantitative analysis of pMoe and total Moesin intensity**. Images from fixed tissues were acquired using an LSM 700 microscope (Zeiss) equipped with a ×63/1.4 Plan Apochromat DIC oil immersion objective. Quantifications of pMoe or total Moesin fluorescence intensities were performed at the periphery of clusters as described in ref. [41]. Briefly, the mean of three z-stacks were used for quantification. To normalize for varying staining between egg chambers, peripheral signals were divided by the intensity measured on nurse cell membranes.

**Quantitative analysis of pMLC2 intensity**. Images from fixed tissues were acquired using an LSM 700 microscope (Zeiss) equipped with a ×63/1.4 Plan Apochromat DIC oil immersion objective. Quantifications of pMLC2 fluorescence intensities at the back, the front, and the side of the cluster were performed as previously described[6].

For more precise analysis of pMLC2 localization and for the line scan analysis, we acquired images from fixed tissues with the Airyscan detector of a Zeiss LSM880 confocal equipped with a ×63/1.4 Plan Apochromat oil immersion objective. Line scans were performed on the Zen Blue software and plotted in GraphPad Prism. Specific line scan were chosen according to the base of the protrusion. The protrusion bases were determined manually by its neck shape (opposite negative

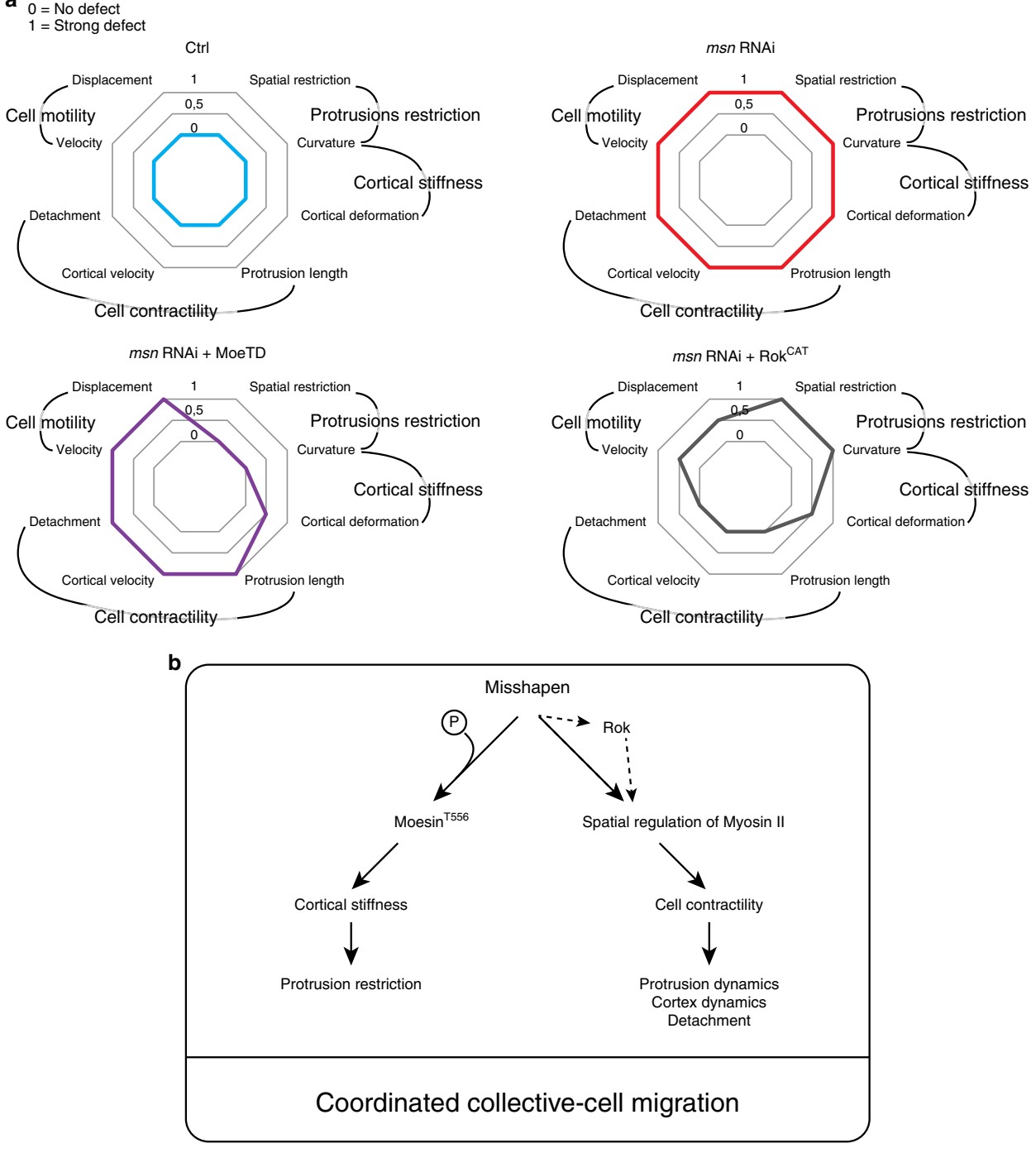

**Fig. 8 Model. a** Schematic representation of the phenotypes observed in control conditions, after depletion of Msn or in Msn-depleted clusters expressing *Moe*[T556D] or *Rok*[CAT]. Phenotypes were scored on a scale of 0 to 1, where 0 is the wild-type phenotype and 1 stands for a strong defects observed after Msn depletion. We categorized each readout into four main categories being Protrusion restriction, Cortical tension, Cell contractility, and Cell Motility. Active Moe (Moe[T556D]) rescues parameters associated with cell communication and partially rescues Msn depletion-induced cell shape defects. Conversely, *Rok*[CAT] expression rescues cell contractility and partially restores cell motility but not cell communication. **b** Working model. The kinase Misshapen regulates cortical stiffness and protrusion restriction by directly phosphorylating Moesin. Independently, Misshapen regulates cluster detachment and dynamics through acto-myosin contractility regulation

curvature shapes). The signal were normalized to the maximal signal of pSqh intensity.

**Tissue staining and antibodies**. Egg chambers were dissected from adult young female in PBS 1X and incubated for 20 min in 4% paraformaldehyde diluted in PBS. Egg chambers were stained using standard immunofluorescence protocol[41]. The following antibodies were used at the indicated dilutions: rabbit polyclonal

anti-phospho-Ezrin (Thr[567])/Radixin (Thr[564])/Moesin (Thr[558]) at 1:100 (Cell Signaling Technology, #3141), rabbit polyclonal anti-phospho-Myosin Light Chain 2 (Ser19) at 1:100 (Cell Signaling Technology, #3671). The chicken polyclonal anti-Moesin was generated for this study (with Immune BioSolutions) and used at a 1:100 dilution. Goat anti-rabbit conjugated to Alexa Fluor 555 (Cell Signaling Technology #4413) and Goat anti-chicken conjugated to Alexa Fluor 555 (Invitrogen #A21437) were used as secondary antibodies at 1:250. Alexa Fluor 555-labeled Phalloidin (Invitrogen, #A34055) was used at 1:250 to visualize F-actin and

DAPI (Sigma-Aldrich, #D9542) was used to stain DNA at 1:10,000. Egg chambers were mounted in VectaShield (Biolynx, #H-1000). Images were processed using Zen software from Zeiss, using only the "level" functions.

**Kinase assay and immunoprecipitation**. For immunoprecipitation of HA-Msn (WT, K61R, D160N), S2 cells (Thermo Fisher #R69007)were washed with cold phosphate-buffered saline (PBS) and lysed in RIPA (20 mM Tris pH 8, 137 mM NaCl, 1% NP-40, 0.1% SDS, 0.5% sodium deoxycholate). Cell lysates were incubated with anti-HA antibodies for 2 h at 4 °C, followed by a 1 h incubation at 4 °C with protein A–Sepharose CL-4B beads (GE Healthcare) and washed with lysis buffer (three times) then with the kinase buffer (one time).

Immunoprecipitated HA-Msn (WT, K61R, D160N) were incubated with 1 μg of purified Moesin CERMAD domain in a kinase buffer (25 mM Tris pH 7.4, 10 mM $MgCl_2$, 5 mM β-glycerophosphate) with 5 μCi of [γ-$^{32}$P] ATP for a total of 10 min at 30 °C. Samples were then separated by SDS–PAGE and radioactive $^{32}$P incorporation was revealed using a PhosphorImager. Reactions were analyzed by autoradiography, Western blots, and Coomassie (total protein).

**Statistical analysis**. Statistical analyses for two unpaired data sets were performed using unpaired Student's $t$-test (Figs. 1h, 2a, 3b, c, f, 4d, f, g, 5b, f, and Fig. 6f). For more than two unpaired data sets we used one-way ANOVA test coupled with Bonferroni correction methods (Figs. 1c, 7d, e, g, h, i, k, Supplementary Figs. 1b, c, 2b, and 4a, b, e). For paired experiments we used two ways ANOVA test (Figs. 2c, e and 6e).

The criterion for all the statistical significance was $p < 0.05$. In figures, mean values were quoted ± s.e.m. Student's $t$-test and one-way ANOVA/Bonferroni tests were performed using GraphPad Prism. Two ways ANOVA tests were performed using SigmaStat software.

## Data availability
All data are available upon reasonable request to the corresponding author.

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

## Acknowledgements

We thank the Bloomington Stock Collection and the Vienna Drosophila RNAi Center for fly stocks. We thank N. Iannantuono for critical reading of the manuscript and C. Charbonneau for technical assistance and helpful discussions. This work was supported by grants from the Canadian Institute for Health Research (CIHR) to G.E. (MOP-148560), P.P.R. (MOP-142374), and to S.C. (MOP-133683) and from the Natural Sciences and Engineering Research Council of Canada (G.E., P.P.R., and S.C.). P.P.R. is scholar of the Fonds de la Recherche du Québec–Santé (FRQS). C.P. and C.Z. received fellowships from the FRQS.

## Author contributions

C.P., S.K. and G.E. designed the project. C.P., S.K., C.Z., L.E.A.D., B.D., P.P.R. conducted the experiments. C.P., S.K., C.Z., L.E.A.D., P.P.R., S.C. and G.E. analyzed the data. C.P., S.K. and G.E. wrote the paper.

## Additional information

**Competing interests:** The authors declare no competing interests.

**Peer Review Information**: *Nature Communications* thanks the anonymous reviewers for their contribution to the peer review of this work. Peer reviewer reports are available.

