## [Peer Review File · Nature Communications]

Reviewers' comments:

Reviewer #1 (Remarks to the Author):

Nature Communication NCOMMS-18-16985-T

"Misshapen coordinates protrusion restriction and actomyosin contractility during collective cell migration" by Plutoni et al.

The border cell cluster is a defined group of cells that undergoes collective migration in *Drosophila* ovaries and serves as an established model for studying mechanisms of cell migration. Formation of a cellular protrusion by a leading cell has to be coordinated with the retraction of the lagging end to enable migration of a cell cluster. How this is accomplished is the question that the authors address in their study. The authors identify the Ste20-like kinase Misshapen as being instrumental in (1) limiting the generation of a force-generating protrusion to the leading cell of the cluster by regulating membrane tension through localized phosphorylation of Moesin, and (2) regulating actomyosin contractility across the cluster (by an unknown mechanism) to ensure that the extension at the front is accompanied by a retraction at the rear end of the cluster.

Summary of the data:

1. The authors previously identified Rab11 and Moesin as important factors for restricting Rac activity and thereby protrusion-generating activity to the leading cell of the border cell cluster. Here, they conducted a screen to identify the kinase that is responsible for phosphorylation for Moesin, and demonstrated biochemically that Misshapen can phosphorylate Moesin, and that localized presence of phosphorylated Moesin along the periphery of border cell cluster and polarized Rac activity depend on Misshapen. The authors' analysis supports a model, in which Rab11 controls the localization of Misshapen via an exocyst-driven transport, which in turn limits phosphorylation of Moesin to the cluster periphery, limiting Rac activity to the leading cell.
2. Second, the authors discovered that Misshapen is also important for coordinated extension and retraction cycles of the cell cluster and the speed of these cycles. This effect seems to be mediated by the regulation of actomyosin contractility by Misshapen, although the molecular mechanism of the interaction remains unknown.
3. The authors demonstrate that the effects on Moesin activity and actomyosin activity are independent functions of the Misshapen kinase that are both instrumental for collective cell migration.

This study gives interesting new insights into the mechanics of collective cell migration and the function of the kinase Misshapen in regulating protrusion behavior and the coordination of leading end extension and lagging end retraction of a migrating cell cluster. This study should be interesting to a wide audience of readers.

In my opinion, the study has been a carefully conducted both at the qualitative and quantitative level, including a careful genetic and biochemical analysis and cutting edge live imaging analysis. The manuscript is transparently written, and the data are clearly documented in figures, graphs, blots and videos. The statistical analysis is based on good sample sizes. My recommendations for improving the manuscript are indicated below.

Critique points:

- 1) Lines 87/88: The authors conclude that Msn is 'required' for phosphorylation of moesin in BCs. Fig 1C shows a significant reduction but not complete of abolishment of pMoe after msn RNAi knock down. The RNAi knock down might not be complete (possibly corroborated by the apparent dominant-negative effect of kinase-dead Msn in Fig 1e) or other kinases might contribute to pMoe levels. It would be safer to speak of an 'important contribution' of Msn to pMoe levels.

2) Along the same lines: Figure 1C suggests a weak (although not significant) effect of other kinases on pMoe levels in BCs. I wonder whether the authors tested any combinations of those kinase knock downs (with/without msn) to see whether this would elicit a stronger reduction of pMoe? Also, Moesin RNAi would have been good control in this experiment.

3) Fig. 1F: Although the other images of pMoe and Msn:YFP distribution of Fig 1 make clear points, I am not convinced that Fig1f shows a 'co-localization' of the two molecules. A more cautious interpretation would be more adequate.

4) A schematic of the border cell cluster with proper labeling of cells and cell interfaces would be very helpful for the reader. Please, explain "periphery of the cluster". Also, it is confusing when the authors speak of 'periphery of BCs' when they probably mean the periphery of the cluster. It might be clearer to systematically speak of BC-nurse cell and BC-BC interfaces.

5) The authors have two graphs (Fig. 2e, 3b) showing numbers of protrusions. What is the difference? Also, it would be helpful if the authors clearly indicated in the figures whether they count large protrusions or any protrusions.

6) The authors infer from their detailed membrane deformation measurements that tension is reduced when Msn is depleted. This approach seems, however, basically a fancy way of revealing membrane protrusions. Whether a mechanism that causes ectopic membrane extensions causes lower tension or lower tension allows additional protrusions seems to remain a chicken-egg problem. Therefore, I suggest to change Lines 166/167 from: "increase of NPN patterns suggests that the depletion of Msn reduces cortical tension" to: 'increase of NPN patterns is consistent with reduce cortical tension....'

7) Lines 201-206: An elongation of protrusions is not necessarily caused by a defect in protrusion contractility but might have other reasons. The conducted experiment does not address the given question properly. However, a similar phenotype has been observed for Myosin II depleted BC clusters (Aranjuez et al.), which would be useful to mention.

8) Fig 6d: dotted lines are not clearly distinguishable from solid lines

9) Material and Methods: Describe Sphericity measurement or provide Reference

10) 2nd paragraph of the discussion is poorly worded

11) Line 312: The authors conclude that Msn has a function in cell-cell communication for which they provide no experimental evidence, however.

12) I recommend including the Model (Suppl Fig 4) into a main figure

Minor points:

- Line 24: the term 'master regulator' sounds exaggerated
- Lines 43 and 197: add Aranjuez ...and McDonald 2016 reference
- Lines 484-486: order of panels in figure legend and text do not agree!

Wording, grammar and typos: suggested changes:

- Line 35: of a small cluster of tightly attached cells.
- 36: follicular epithelium (change throughout manuscript)
- 48: formation is largely limited
- 51: Moesin loss-of-function conditions
- 53: opposing traction or pulling forces (??)
- 83: msn-RNAi constructs

- 87: not affect pMoe at a significant level
- 92: kinase-inactive Msn proteins/isoforms
- 98: inactive mutant form of Msn
- 113: redistributed from the periphery of the cluster to borders between BCs
- 125: and usually form a single prominent protrusion
- 127: extend several prominent protrusions (??)
- 141: control clusters
- 143: revise sentence, e.g: Hence, while long protrusions (>9 μm) are restricted to the leader cell and point in the direction of the migration in control clusters, long protrusion are formed by multiple BCs and point in different directions when Msn is depleted.
- 148: activity to the front cell
- 150: to the leader cell
- 168: control clusters
- 178: protrusion dynamics
- 189: but affects actomyosin organization in
- 235: protrusion size
- 273: Msn was found to be involved
- 277-289: revise text
- 294: Moesin by Slik occurs
- 305: important roles/functions
- 306: shown to be involved
- 314-315: in accordance with a function of Msn in Moesin activation. When we analyzed
- 320: where it phosphorylates Moesin
- 322: activity of guidance receptor tyrosine kinases
- 324: effect on Moesin seemed epistatic (??)
- 326; cellular borders insidepresent at these borders
- 331: LOK acts as a wedge
- 333: cell borders of the cluster
- 353: form extensions at the front and contract them at the back.
- 365: E-cadherin-based cell-cell, thereby helping
- 368: recent studies have identified a function for Msn
- 412: to migrate and still have not detached by stage 10.
- 420: Analysis of the orientation of protrusions longer than 9 μm
- 424: for each condition
- 440: while others
- 462: dotted circle
- 473: each time point
- 483: 9 μm
- 487: each condition
- 488: each time point
- 499: at the cell-cell border
- 520: rescued by either constitutively active forms
- 528: partially restores
- 572: Dresden (?)

Reviewer #2 (Remarks to the Author):

The manuscript 'Misshapen coordinates protrusion restriction and actomyosin contractility during collective cell migration' by Plutoni et al, 2018 investigates the mechanism by which Ste20 kinase

Misshapen phosphorylates Moesin and regulates a protrusion restriction mechanism. In addition to phosphorylating Moesin, Misshapen also promotes cell contractility through Rok and Myosin II in order to regulate cluster detachment and protrusion dynamics. This study makes significant advancement in the field since the last study was published involving roles of misshapen in border cells (Cobrerros-Reguera, 2010) and furthers their previous investigation on the role of Moesin in border cells (Ramel et al., 2013). The manuscript is well written and the data presented here are convincing for the most part. The movies are beautiful. If the authors can address the questions and comments below satisfactorily, the work would be of interest to the readers of Nature Communications.

1. The use of the term master regulator in the abstract is inappropriate and an unnecessary overstatement. A master regulator is a molecule that is necessary and sufficient for a particular developmental event and should be relatively specific for that event. The term was invented to describe the striking effect of Eyeless on eye development. Not only does loss of eyeless results in complete loss of eye development but ectopic eyeless is sufficient to produce ectopic eyes. Nothing about Misshapen fits this definition.

2. The msn migration defects are much stronger than the moesin migration phenotype. In fact in the original work on msn, it looked as if msn might prevent border cell specification. Moesin k.d. causes only mild perturbation of migration. Can the authors address this and indicate whether it is likely that Msn has additional targets?

3. Statistical analysis: The only listed statistical method is an unpaired student's t-test for two data sets. This may be either incomplete, inappropriate, or not clearly worded. For example, in Figure 1C there is a comparison of intensity in multiple RNAi conditions compared to the control, which is a multiple comparisons situation. To account for this a correction method should be applied (Bonferroni or other) or an explanation given for why correction is not done in this case. (Also true for Figure 7 D/H/I/K).

4. Fig1G and I: What is the distribution of Msn in the Rab11S25N induced ectopic protrusions?

Fig1J: Is it possible to show sec15:mCherry, Rab11 and Msn:YFP in the same vesicular structure? This would make it more convincing that Msn is transported via Rab11 through sec15.

Fig1-H: It is not immediately clear how the intensity ratio is being measured or what one N consists of. Does N=15 mean all contacts within 15 border cells, or 15 BC/NC and 15 BC/BC contacts within any number of border cells. How many egg chambers from how many flies from how many separate experiments were analysed? BC/NC ratio can be measured by just quantifying all BC-BC contacts and then all BC-NC contacts and taking a ratio. But how is a ratio determined for BC/BC membranes? Which membrane is divided by which, how many total membranes are measured, and what does it mean to have a ratio of 2 in the Rab11S25N situation (or even any ratio other than 1 for BC/BC contact)? Is this a ratio of Msn::YFP over some other marker/protein?

5. Fig2-E: Statistical analysis. If this is showing combined data of number of protrusions at every single individual time point in multiple different movies, an unpaired t-test for two samples is not appropriate, as the sampling is repeated and not independent within each movie. It would be necessary to take a linear/mixed modeling approach, re-sampling/dependent t-test, or a restructuring of the data depending on how exactly the experiment was carried out.

6. Fig3-F: "For quantification, we used five different stacks for each cluster". It is not clear what this means. The figure legend states n=10 for each condition. Does this mean that there are 50 individual data points for each condition? How were the planes chosen? It looks like there are more data points for msn than for control. More importantly this causes the same issue as mentioned above, if the different planes from the same cluster are being measured then a student's t-test is not appropriate, as the samples are no longer independent, and a different test must be used.

Fig3-F: What are the specific, unbiased criteria for what constitutes front vs. back of the cluster?

7. Fig5B: Local pMLC enrichment is shown for side and back of the cluster and this doesn't change. Does this only include side membrane or also in msnRNAi side protrusions? If local pMLC in side protrusions is not changed, is there a different proposed mode of contractility compared to lead front protrusions?

Fig5D: Authors say in line 191 'Furthermore, Myosin II is known to form transient foci at the BCs periphery (Fig. 5c, black arrows), which are necessary for cluster contraction [21, 22]. We found that these foci are absent in Msn-depleted clusters (Fig. 5d).'

Fig5-E: It is not clear how these specific line scans were chosen. In the control, is the neck defined by the opposing strong pMLC2 signal? If so how is the neck defined in the msn RNAi, where it lacks strong pMLC2 signal?

8. Fig6-E/F: Not all samples are independent, since sampling multiple time points from movies, violating the assumptions of a t-test.

9. Fig7: "...RokCat...and found that it rescues the Msn depletion-induced...global morphodynamics defects". It is not clear comparing the morphodynamic chart from this figure with the control from 6B that there has been a rescue of the phenotype. It also isn't clear why this would be expected to be the case, as the effects on Moesin due to Msn knockdown should still cause abnormalities in the morphodynamic map. Could you clarify what exactly is considered rescue of morphodynamic defects in this figure?

10. Discussion – Line 287: Were complementation experiments with Ecad mutants done but not mentioned in the rest of the text?

Minor Comments:

F2-E and F3-B: Is there an intended difference in what we should take away from these two figures, or is it essentially the same data with the same conclusion?

F4: Was curvature analysis done on a single plane or a maximum intensity stack? If on a single plane how is that plane chosen?

F7: There were genetic issues with combining RNAi #1 with Rok-Cat, but why could the experiments not all be done using RNAi #2? When trying to confirm that Msn is truly independently regulating two pathways it would be reassuring to see the experiments done in the same RNAi background.

Methods: Time of incubation for RNAi lines should be included in addition to the temperature (it is currently present under Ste20 kinase screen section, but not under the Drosophila genetics section)

Methods: Under Quantitative analysis of pMoe and total Moesin intensity it is mentioned that intensities were taken from three z-stacks. For clarity, it should be made explicit if this is the sum of the three z-stacks, as well as what the total size is in microns.

Figure6: Talks about extension and retraction. However the sides of cluster also have contractility. How is that affected? A quantification for the same should also be added.

F5-E and S2: Not clear how the base of protrusion was chosen. Protrusion length should be marked and tip and base should be defined. Also how the signal is normalized?

Fig7: Rok-CAT control phenotypes should be added here. Does the protrusion lifetime also change in each of these manipulations?

Supplementary figure S4: is not labeled. If possible please incorporate the model in main text. Also the model is little hard to absorb. Here the authors show that Rok and Myosin II regulate cell contractility and thus regulate protrusion dynamics and detachment. Moesin dependent regulation of cortical tension governs the protrusion restriction phenomenon. However many previous studies (including Aranjuez et al, 2016) argue that F-actin and MyoII generate cortical tension. It is not very well discussed in the manuscript that how Moesin dependent cortical tension but not the Myosin II dependent cortical tension regulates protrusion restriction.

defects in place of 'defaults' in line 260.

In Supplementary Fig S4 b (not labeled) in the coordinated collective cell migration model, 'protrusion dynamic' should be changed to protrusion dynamics and 'cortex dynamic' should be changed to cortex dynamics.

Reviewer #3 (Remarks to the Author):

The manuscript by Plutoni et al. addresses the mechanisms of collective cell migration by studying border cell migration in *Drosophila*. A previous report (Cobrerros-Reguera, 2010) showed that Msn is essential for border cell migration. These authors provided evidence that Msn acts by modulating the levels or localization of E-cadherin. As Ezrin-Radixin-Moesin (ERM) proteins are known substrates of Ste-20 like kinases (like Msn) Cobrerros-Reguera also tested whether Moesin is required for border cell migration, but did not find evidence for this. Plutoni et al. now claim that the Ste20-like kinase Msn phosphorylates Moesin in border cells and that Msn and Moesin are required for restricting protrusion formation to the leading cells of the migrating cluster. Moreover, they claim that Msn is critical for generating contractile forces at the rear of the cluster. Rescue experiments indicate that Msn acts via a Moesin-dependent and a second, Moesin-independent pathway. The authors conclude that Msn coordinates protrusion formation at the leader cells with the induction of contractile forces by regulating two independent pathways.

One novel and interesting aspect of the study is that the authors identify Msn as a kinase that phosphorylates Moesin in the context of border cell migration. However, I find some of the major claims not well supported. The authors claim that Msn is a master regulator. However, since Msn (and P-Moesin) is uniformly located along the periphery of the border cell cluster, it remains unclear whether Msn has an instructive role during border cell migration (distinguishing front from rear). Moreover, the authors claim that Moesin regulates contractile forces, but it seems equally possible based on the authors' data that Moesin is required for cortical stiffness (as shown previously in a different context, e.g. Kunda et al., 2008). Moreover, on the mechanistic level, it remains unclear how Msn activates Moesin.

Specific comments

-Fig. 1. The authors claim that Msn is required for Moesin phosphorylation and border cell migration based on two RNAi lines. Since RNAi can have unspecific effects, the authors should use available msn alleles to corroborate their findings.

-Fig. 1. The authors show that Msn phosphorylates the CERMAD domain of Moesin. Does Msn phosphorylate Thr556, which is required for unfolding and activation of Moesin?

-Fig. 1e. The authors claim that wild-type, but not a kinase-dead version of Msn rescues msn-RNAi flies. As a control, the authors should test whether Msn-K160D is expressed at a comparable level as the wild-type Msn protein. Moreover, it appears that Msn-K160D enhances the border cell migration defects of msn-RNAi flies. Do the authors think that Msn-K160D acts as a dominant-negative?

- Fig. 1. The authors claim that their data shows that Msn Phosphorylates Moesin to promote border cell migration. This conclusion would be much strengthened if the authors could show that an unphosphorylatable form of Moesin would fail to rescue moesin deficiency in border cells.

-Fig. 1. P-Moesin is not only present at the periphery of the border cell cluster, but also on the membranes of neighboring cells (nurse cells). In the authors' experiments, Msn is depleted in the border cells resulting in a reduction of P-Moesin in these cells. Does Msn specifically phosphorylate Moesin in the border cell cluster, or is this phosphorylation more widespread (incl. the nurse cell)?

-Fig. 4a. The authors state that "As moesin generates cortical tension in its active state [10].... My reading of reference 10 is that Kunda et al. 2008 (Ref. 10) in fact used AFM to measure cortical stiffness (Young's modulus), which in mitotic cells depends on moesin. Moreover, a phosphomimetic variant, T559D- Moesin induced a >2-fold increase in cortical rigidity. This increased cortical rigidity was independent of Myosin activity.

-Fig. 4b-d. The analysis of membrane-curvature probably reflects the formation of protrusions. It unclear how the authors infer from this that Msn regulates cortical tension. Membrane curvature likely depends on tension, but also on cortical stiffness. The authors need to distinguish between tension and stiffness.

-Fig. 4e-g. The authors conclude that Msn regulates protrusion dynamics, in addition to overall cortical tension around the cluster. To this reviewer, it would be more plausible to say that Msn-dependent cortical tension at the cluster periphery restricts protrusion formation. What makes the authors conclude that Msn has two distinct functions here?

-Fig. 5e. How reproducible is the symmetric distribution of pMLC2 in the wild-type and its absence in the msn depleted situation? The authors seem to show only a single border cell cluster.

-Fig. 5. Panel 'f' seems to be missing.

-Fig. S2. The authors claim that Msn localizes to cortical actin-rich structures including at the base and the tip of protrusions. However, there are many Life-Act spots that do not co-localize with Msn. Moreover, the authors speculate that Msn may regulate Myosin II activity at these spots. Is there any evidence for this?

-Fig. 6b. The authors claim that extension and contraction events are coordinated across the cluster. However, if the border cell cluster almost constantly retracts back (as the authors state in the text (line 216)), I do not see how these two events are coordinated.

-Fig. 6c. This panel is confusing. In Fig.6b, the authors show that both the front and the back of a border cell cluster can extend and retract. Fig. 6c now shows velocities of extension and retraction. Do these velocities relate to the front/back end of the cluster?

Comments to the reviewers:

We thank the reviewers for their positive comments and constructive suggestions. We were pleased to read that the reviewers thought that our work is of high interest and we think that their requests and suggestions have significantly improved our manuscript. Here we submit a revised version where the changes are highlighted. The major changes are:

- We clarified the relationship between stiffness and protrusion restriction. Actually, in the initial manuscript we misused the word “tension” to describe “stiffness”. This is now corrected in the manuscript, and we rephrase several sentences in the results and discussion sections regarding the link between stiffness and protrusion restriction.
- We added the various controls asked by the reviewers, including: We show that Misshapen phosphorylates the right residue on Moesin (Thr556). We duplicated most of the analysis of our complementation assay with the second RNAi line. We show that Misshapen co-localizes on vesicles with both Rab11 and Sec15. We determined that non-phosphorylatable Moesin (MoeTA) is unable to rescue any phenotype induced by the loss of Misshapen.
- We corrected the statistical analysis when appropriate, and increased the sample pool when required.

Here are specific responses to the reviewers’ comments (Figures for reviewers are at the end of this document).

Reviewer #1:

[...]

This study gives interesting new insights into the mechanics of collective cell migration and the function of the kinase Misshapen in regulating protrusion behavior and the coordination of leading end extension and lagging end retraction of a migrating cell cluster. This study should be interesting to a wide audience of readers.

In my opinion, the study has been a carefully conducted both at the qualitative and quantitative level, including a careful genetic and biochemical analysis and cutting edge live imaging analysis. The manuscript is transparently written, and the data are clearly documented in figures, graphs, blots and videos. The statistical analysis is based on good sample sizes. My recommendations for improving the manuscript are indicated below.

We thank the reviewer for the positive assessment of our manuscript.

Critique points:

1) Lines 87/88: The authors conclude that Msn is 'required' for phosphorylation of moesin in BCs. Fig 1C shows a significant reduction but not complete abolishment of pMoe after msn RNAi knock down. The RNAi knock down might not be complete (possibly corroborated by the apparent dominant-negative effect of kinase-dead Msn in Fig 1e) or other kinases might contribute to pMoe levels. It would be safer to speak of an 'important contribution' of Msn to pMoe levels.

We have changed the manuscript accordingly.

Line 83-86. "While depletion of Tao resulted in a minor decrease of pMoe staining, depletion of Pak3 and Slik did not significantly affect pMoe levels (Fig. 1c). Overall, this demonstrates that Msn is essential for the normal phosphorylation of Moesin in BCs."

Note that Moesin is also phosphorylated in the nurse cells that surround the cluster and hence we were not expecting a complete loss of pMoe.

2) Along the same lines: Figure 1C suggests a weak (although not significant) effect of other kinases on pMoe levels in BCs. I wonder whether the authors tested any combinations of those kinase knock downs (with/without msn) to see whether this would elicit a stronger reduction of pMoe? Also, Moesin RNAi would have been good control in this experiment.

Recent work showed that Tao, which is the only other kinase that has a significant effect, is an upstream activator of Msn in other tissues [1] and we are currently investigating if it also acts upstream of Msn in border cells. We have characterized Moesin RNAi in a previous publication [2]. Moesin is a very stable protein. For example, 6 days dsRNA treatment is necessary to obtain a strong reduction of protein levels in S2 cells (see Figure for Reviewer #1) and null mutants are viable up to late pupal stages. Moesin loss of function is thus difficult to study in border cells.

3) Fig. 1F: Although the other images of pMoe and Msn:YFP distribution of Fig 1 make clear points, I am not convinced that Fig1f shows a 'co-localization' of the two molecules. A more cautious interpretation would be more adequate.

Co-localization is a word of art. In our understanding, it can cover everything from a complete overlap of the distribution of two proteins to a very partial overlap. Hence, we agree that we should have been more precise and have changed the text to:

Line 107-108. "We found that Msn and Moesin co-localized in specific regions of the peripheral cortex of the cluster (Fig. 1f, arrows).

4) A schematic of the border cell cluster with proper labeling of cells and cell interfaces would be very helpful for the reader. Please, explain "periphery of the cluster". Also, it is confusing when the authors speak of 'periphery of BCs' when they probably

mean the periphery of the cluster. It might be clearer to systematically speak of BC-nurse cell and BC-BC interfaces.

We have added a scheme in Fig.1h, and we thank the reviewer to have pointed to our mistake: indeed periphery of BCs means periphery of the cluster. All the instances were changed accordingly.

5) The authors have two graphs (Fig. 2e, 3b) showing numbers of protrusions. What is the difference? Also, it would be helpful if the authors clearly indicated in the figures whether they count large protrusions or any protrusions.

The quantifications were done with different data sets, Fig. 2e was quantified from time-lapse recordings, whereas Fig.3b was quantified from fixed tissues, allowing for a higher spatial resolution. We have now clarified this in the figure and in the legends.

Lines 409-412 and 420-421.

6) The authors infer from their detailed membrane deformation measurements that tension is reduced when Msn is depleted. This approach seems, however, basically a fancy way of revealing membrane protrusions. Whether a mechanism that causes ectopic membrane extensions causes lower tension or lower tension allows additional protrusions seems to remain a chicken-egg problem. Therefore, I suggest to change Lines 166/167 from: "increase of NPN patterns suggests that the depletion of Msn reduces cortical tension" to: 'increase of NPN patterns is consistent with reduce cortical tension....'

Line169-170. We modified the sentence accordingly. "This increase of the number of NPN patterns is consistent with reduced cortical stiffness."

Note that in agreement with reviewer #3 comments, we also changed "tension" with "stiffness", as what curvature is actually a proxy for membrane stiffness.

7) Lines 201-206: An elongation of protrusions is not necessarily caused by a defect in protrusion contractility but might have other reasons. The conducted experiment does not address the given question properly. However, a similar phenotype has been observed for Myosin II depleted BC clusters (Aranjuez et al.), which would be useful to mention.

We added the reference to Aranjuez et al. at line 205 and reformulated lines 202-205. This specific association with protrusions suggests that, in addition to potentially generating cortical stiffness via Moesin activation, Msn could also regulate protrusion retraction. Interestingly, a similar phenotype has been observed for Myosin II depleted BC clusters [7]. To test if the dysregulation of Myosin II affects the morphology of

protrusions, we quantified the distance from the nuclei to the tip of the protrusion and found an elongation of both front and side protrusions after Msn depletion (Fig. 5g).

8) Fig 6d: dotted lines are not clearly distinguishable from solid lines

We have changed the pattern of lines in Fig.6d.

9) Material and Methods: Describe Sphericity measurement or provide Reference

Line 660-664. We have added the description of the measurement of sphericity in the material and methods. 3D reconstructions, representations, cluster sphericity and protrusion length measurements were performed in Imaris (Bitplane) from 3D volume reconstructions of LifeAct::GFP and DAPI, both acquired with the Airyscan detector of a Zeiss LSM880 confocal. Sphericity was measured in the statistical analysis module of Imaris.”

10) 2nd paragraph of the discussion is poorly worded

We rewrote this paragraph. Line 277-290.

11) Line 312: The authors conclude that Msn has a function in cell-cell communication for which they provide no experimental evidence, however.

We have removed this sentence.

12) I recommend including the Model (Suppl Fig 4) into a main figure

We have moved the model to Fig. 8.

Minor points:

- Line 24: the term 'master regulator' sounds exaggerated
- Lines 43 and 197: add Aranjuez ...and McDonald 2016 reference
- Lines 484-486: order of panels in figure legend and text do not agree!

Wording, grammar and typos: suggested changes:

- Line 35: of a small cluster of tightly attached cells.

- 36: follicular epithelium (change throughout manuscript)
- 48: formation is largely limited
- 51: Moesin loss-of-function conditions
- 53: opposing traction or pulling forces (??)
- 83: msn-RNAi constructs
- 87: not affect pMoe at a significant level
- 92: kinase-inactive Msn proteins/isoforms
- 98: inactive mutant form of Msn
- 113: redistributed from the periphery of the cluster to borders between BCs
- 125: and usually form a single prominent protrusion
- 127: extend several prominent protrusions (??)
- 141: control clusters
- 143: revise sentence, e.g: Hence, while long protrusions (>9 μm) are restricted to the leader cell and point in the direction of the migration in control clusters, long protrusion are formed by multiple BCs and point in different directions when Msn is depleted.
- 148: activity to the front cell
- 150: to the leader cell
- 168: control clusters
- 178: protrusion dynamics
- 189: but affects actomyosin organization in
- 235: protrusion size
- 273: Msn was found to be involved
- 277-289: revise text
- 294: Moesin by Slik occurs
- 305: important roles/functions
- 306: shown to be involved
- 314-315: in accordance with a function of Msn in Moesin activation. When we analyzed
- 320: where it phosphorylates Moesin
- 322: activity of guidance receptor tyrosine kinases
- 324: effect on Moesin seemed epistatic (??)
- 326; cellular borders insidepresent at these borders

- 331: LOK acts as a wedge
- 333: cell borders of the cluster
- 353: form extensions at the front and contract them at the back.
- 365: E-cadherin-based cell-cell, thereby helping
- 368: recent studies have identified a function for Msn
- 412: to migrate and still have not detached by stage 10.
- 420: Analysis of the orientation of protrusions longer than 9 μm
- 424: for each condition
- 440: while others
- 462: dotted circle
- 473: each time point
- 483: 9 μm
- 487: each condition
- 488: each time point
- 499: at the cell-cell border
- 520: rescued by either constitutively active forms
- 528: partially restores
- 572: Dresden (?)

We have corrected all these comments and thank the reviewer to pointing at these errors.

Reviewer #2:

[...]

This study makes significant advancement in the field since the last study was published involving roles of misshapen in border cells (Cobrerros-Reguera, 2010) and furthers their previous investigation on the role of Moesin in border cells (Ramel et al., 2013). The manuscript is well written and the data presented here are convincing for the most part. The movies are beautiful. If the authors can address the questions and comments below satisfactorily, the work would be of interest to the readers of Nature Communications.

We thank the reviewer for these positive comments.

1. The use of the term master regulator in the abstract is inappropriate and an unnecessary overstatement. A master regulator is a molecule that is necessary and sufficient for a particular developmental event and should be relatively specific for that

event. The term was invented to describe the striking effect of Eyeless on eye development. Not only does loss of eyeless results in complete loss of eye development but ectopic eyeless is sufficient to produce ectopic eyes. Nothing about Misshapen fits this definition.

In the light of these explanations, we definitively agree with the reviewer that our use of “master regulator” is incorrect. We have changed “master regulator” to “key regulator” in the manuscript. Line 24.

2. The msn migration defects are much stronger than the moesin migration phenotype. In fact in the original work on msn, it looked as if msn might prevent border cell specification. Moesin k.d. causes only mild perturbation of migration. Can the authors address this and indicate whether it is likely that Msn has additional targets?

Most of the manuscript and our model precisely highlight that Moesin cannot be the only target of Msn. Indeed, the first part of our manuscript shows that Msn is the Moesin kinase as it fulfill our prediction for a Moesin kinase: 1) its kinase activity is required for BC migration, 2) it can directly phosphorylate Moe in vitro, 3) it localizes at the periphery of the cluster, where Moe is phosphorylated, 4) its localization depends on Rab11, 5) its depletion induces a loss of the Rac/protrusion restriction to leader cell. Then, we acknowledge that we have differences compared to a Moe/Rab11 phenotype and dig into the fact that Msn acts on contractility and Moesin II. Furthermore, we show that this role of Msn is actually independent of its action on Moe, and thus that it has to act on a different target in BCs

Our manuscript, however, did not investigate the potential BC specification phenotype after Msn depletion. We have now determined that, to the difference of mutants, RNAi mediated depletion of Msn does not reduce the number of BCs, most probably because the depletion of Msn induced by c306-GAL4 is effective after BC differentiation. We comment on this in the discussion. Line 282-286.

As mention in the manuscript, we perform our experiments at 25°C so that the effect of depletion is partial and not complete. In these conditions, we can characterize in details the impact of Msn depletion on protrusion restriction and on contractility. More precisions about temperatures were added to the Materials and Methods. Line 592 and 602-603.

Finally, regarding Moesin k.d., we think that while our previous findings (Ramel et al., NCB) are consistent, we might not be entirely depleting Moe due to its persistency (see comments to Reviewer #1 and the Figures for reviewer).

3. Statistical analysis: The only listed statistical method is an unpaired student's t-test for two data sets. This may be either incomplete, inappropriate, or not clearly worded. For example, in Figure 1C there is a comparison of intensity in multiple RNAi conditions compared to the control, which is a multiple comparisons situation. To account for this a correction method should be applied (Bonferroni or other) or an explanation given for why correction is not done in this case. (Also true for Figure 7 D/H/I/K).

We thank the reviewer to pointing to this inconsistency. We have now used a one way ANOVA with Bonferroni correction (for Fig 1c, Fig.7 d,e,h,i and k, Supplemental Fig.1 b and c, Supplemental Fig.2 (new Supplemental Figure) B and Supplemental Fig.3 (now Supplemental Fig.4) a,b and e). The outcome of the statistics analysis remains unchanged.

We added detailed information in the Material and Methods, Statistical analysis. Line 715-723.

4. Fig1G and I: What is the distribution of Msn in the Rab11S25N induced ectopic protrusions?

Msn is diffuse, almost entirely absent from protrusions after expression of Rab11SN, indeed, as highlight in Fig.1G, it is redistributed to internal BC-BC contacts.

Fig1J: Is it possible to show sec15:mCherry, Rab11 and Msn:YFP in the same vesicular structure? This would make it more convincing that Msn is transported via Rab11 through sec15.

Due to restrictions in the different tags available, we did not perform a triple staining. However, we now show that Msn::YFP is localized in both Rab11 positive vesicles and in Sec15 positive vesicles (Supplemental Fig. 3a).

Fig1-H: It is not immediately clear how the intensity ratio is being measured or what one N consists of. Does N=15 mean all contacts within 15 border cells, or 15 BC/NC and 15 BC/BC contacts within any number of border cells. How many egg chambers from how many flies from how many separate experiments were analysed? BC/NC ratio can be measured by just quantifying all BC-BC contacts and then all BC-NC contacts and taking a ratio. But how is a ratio determined for BC/BC membranes? Which membrane is divided by which, how many total membranes are measured, and what does it mean to have a ratio of 2 in the Rab11S25N situation (or even any ratio other than 1 for BC/BC contact)? Is this a ratio of Msn::YFP over some other marker/protein?

We have now added a scheme in Fig.1h, and we clarified both the Fig. 1h and the description in the legend. “**h**. Schematic representation of border cell cluster with labelling of the different cells and cell interfaces. **e**. Quantification of the ratio between the mean Msn::YFP fluorescence signal at the BCs interface within the cluster (BC/BC) or at the periphery of the cluster (BC/NC interface). (n=15 BCs for both condition).” Line 397-400.

n are for the number of clusters that have been analyzed and they were from several flies (usually 10 females).

5. Fig2-E: Statistical analysis. If this is showing combined data of number of protrusions at every single individual time point in multiple different movies, an unpaired t-test for two

samples is not appropriate, as the sampling is repeated and not independent within each movie. It would be necessary to take a linear/mixed modeling approach, re-sampling/dependent t-test, or a restructuring of the data depending on how exactly the experiment was carried out.

We corrected the statistical analysis for combined and paired observations. We performed a two way ANOVA for statistical analysis of Figures 2c,e and Fig. 6 e. We added detailed information in the Material and Methods, Statistical analysis. Line 715-723.

We corrected the figure legends of Fig. 6f. The dot plot shows the means of the velocity variance for each movies. As we compared only means between two unpaired data sets, there is no coupled observations and a student's t test is appropriate. "f. Histogram representing the mean variance for extensions and contractions velocities measured in g. ($n > 7$).". Line 476.

6. Fig3-F: "For quantification, we used five different stacks for each cluster". It is not clear what this means. The figure legend states $n=10$ for each condition. Does this mean that there are 50 individual data points for each condition? How were the planes chosen? It looks like there are more data points for msn than for control. More importantly this causes the same issue as mentioned above, if the different planes from the same cluster are being measured then a student's t-test is not appropriate, as the samples are no longer independent, and a different test must be used.

We apologize that our description was imprecise and confusing. For each cluster, we have used five different planes from a z-stack, the central plane was containing the polar cells. The mean of the different planes was calculated and use as a single data point (for each of the ten clusters analyzed). As such, a student's t-test seems adequate. The number of data points is actually identical between Msn and control. We completed the Figure legends of Fig. 3f. "f. Quantification of the ratio of the FRET indexes measured at front and back of control and Msn-depleted clusters. Dot plots represents the means of five different planes for each cluster ($n=10$ clusters)". Line 426-429.

Fig3-F: What are the specific, unbiased criteria for what constitutes front vs. back of the cluster?

The cluster migrates along antero-posterior axis of the egg chamber, which is used to define the front and the back of the cluster. We used a quadrant on the BC cluster to delineate the front and the back. A scheme is now inserted in figure 3f to clarify our measurement.

7. Fig5B: Local pMLC enrichment is shown for side and back of the cluster and this doesn't change. Does this only include side membrane or also in msnRNAi side protrusions? If local pMLC in side protrusions is not changed, is there a different proposed mode of contractility compared to lead front protrusions?

Fig. 5b explicitly contained the quantifications of pMLC2 at the side, including protrusions. No differences were seen regarding the mean intensities measured, however as shown in the following panels, the discrete distribution of pMLC2 changes both in front protrusion and at the periphery.

It is difficult to assess the effect of Msn depletion on pMLC2 in side protrusions since every protrusion is morphologically identical in this condition. The idea that contractility is differently regulated in side protrusions is an interesting hypothesis that we have not investigated this here.

Fig5D: Authors say in line 191 'Furthermore, Myosin II is known to form transient foci at the BCs periphery (Fig. 5c, black arrows), which are necessary for cluster contraction [21, 22]. We found that these foci are absent in Msn-depleted clusters (Fig. 5d).'

Fig5-E: It is not clear how these specific line scans were chosen. In the control, is the neck defined by the opposing strong pMLC2 signal? If so how is the neck defined in the msn RNAi, where it lacks strong pMLC2 signal?

The neck was defined by its shape, in particular the curvature changes at the cortex. It was determined by using Lifeact and not pMLC2. We completed the Material and Methods. "Specific line scan were chosen according to the base of the protrusion. The protrusion bases were determined manually by its neck shape (opposite negative curvature shapes). The signal were normalized to the maximal signal of pSqh intensity." Line 686-688.

8. Fig6-E/F: Not all samples are independent, since sampling multiple time points from movies, violating the assumptions of a t-test.

We apologize for the mistake and we now changed our statistical analysis of Fig.6e to a two ways ANOVA. For Fig.6f, however, we have a single variance per movie analyzed and hence a t-test seems valid to us as each movie is independent / not paired. We added detailed information in the Material and Methods, Statistical analysis. Line 715-723.

9. Fig7: "...RokCat...and found that it rescues the Msn depletion-induced...global morphodynamics defects". It is not clear comparing the morphodynamic chart from this figure with the control from 6B that there has been a rescue of the phenotype. It also isn't clear why this would be expected to be the case, as the effects on Moesin due to Msn knockdown should still cause abnormalities in the morphodynamic map. Could you clarify what exactly is considered rescue of morphodynamic defects in this figure?

We thank the reviewer to pointing our lack of precision. We now describe that the rescue concerns the global contractility, and the contraction and extension velocities. Line 255-256.

As Rok^{CAT} is expressed uniformly in the cells, contractility seems not restricted to specific domains (punctae and protrusions), which explains why the morphodynamic maps are not identical to controls.

10. Discussion – Line 287: Were complementation experiments with Ecad mutants done but not mentioned in the rest of the text?

No, we did not perform complementation experiments with E-Cadherin. Previous work [3] shows that E-Cadherin is mislocalized in *msn* loss-of-function, but not reduced. We anticipate that expressing E-Cadherin would not rescue any *msn* phenotype.

Minor Comments:

F2-E and F3-B: Is there an intended difference in what we should take away from these two figures, or is it essentially the same data with the same conclusion?

No, but they were performed on different conditions (fixed vs live imaging). This has been clarified in the figure and in figure legends. Lines 409-412 and 420-421.

F4: Was curvature analysis done on a single plane or a maximum intensity stack? If on a single plane how is that plane chosen?

On a single plane. We added more detail on these experiments in the methods. "The analysis was done on a single plane from confocal movies. Three different planes were recorded. From these, we choose the plan with the best signal to noise ratio to perform the analysis." Line 641-643.

F7: There were genetic issues with combining RNAi #1 with Rok-Cat, but why could the experiments not all be done using RNAi #2? When trying to confirm that Msn is truly independently regulating two pathways it would be reassuring to see the experiments done in the same RNAi background.

We have now included experiments with the second RNAi, which are entirely consistent with our previous experiments (Supplemental Fig. 5a,b.). However, we could not repeat all the experiments and quantifications requiring live imaging.

Methods: Time of incubation for RNAi lines should be included in addition to the temperature (it is currently present under Ste20 kinase screen section, but not under the Drosophila genetics section)

The information was added in the main text and in the methods. Line 592 and 602-603.

Methods: Under Quantitative analysis of pMoe and total Moesin intensity it is mentioned that intensities were taken from three z-stacks. For clarity, it should be made explicit if this is the sum of the three z-stacks, as well as what the total size is in microns.

We have added details in the methods. Basically: we did measure the intensity of the signal in three consecutive planes of each z-stacks and showed the mean values. Line 673-676.

Figure6: Talks about extension and retraction. However the sides of cluster also have contractility. How is that affected? A quantification for the same should also be added.

Our analysis exploits the ADAPT plugin that does not allow the segmentation of the cluster periphery in sections. Hence, retractions and extensions are measured all along the periphery. We can only assess the positioning of extensions and retractions through the morphodynamic maps.

F5-E and S2: Not clear how the base of protrusion was chosen. Protrusion length should be marked and tip and base should be defined. Also how the signal is normalized?

The neck was defined by curvature changes at the cortex, determined by using Lifeact. The signal is normalized to the total pSqh intensity. We now describe this in the Material and Methods. Line 686-688.

Fig7: Rok-CAT control phenotypes should be added here. Does the protrusion lifetime also change in each of these manipulations?

We added this experiment in Supplemental Fig. 5c. We also did it for MoeTD alone. In both cases, no detachment and protrusion defect were observed (Supplemental Fig. 5c,d). We performed morphodynamic analysis of Rok-CAT alone, and it does not induce noticeable changes in extension or contraction velocity. (Figure for Reviewer #2, orange column).

Supplementary figure S4: is not labeled. If possible please incorporate the model in main text. Also the model is little hard to absorb. Here the authors show that Rok and Myosin II regulate cell contractility and thus regulate protrusion dynamics and detachment. Moesin dependent regulation of cortical tension governs the protrusion restriction phenomenon. However many previous studies (including Aranjuez et al, 2016) argue that F-actin and MyoII generate cortical tension. It is not very well discussed in the manuscript that how Moesin dependent cortical tension but not the Myosin II dependent cortical tension regulates protrusion restriction.

The model was incorporated in the main text. (Fig. 8).

We have now replaced tension by stiffness as we think that this is more in accordance to the known function of Moesin.

defects in place of 'defaults' in line 260.

In Supplementary Fig S4 b (not labeled) in the coordinated collective cell migration model, 'protrusion dynamic' should be changed to protrusion dynamics and 'cortex dynamic' should be changed to cortex dynamics.

These mistakes were corrected.

Reviewer #3:

[...]

One novel and interesting aspect of the study is that the authors identify Msn as a kinase that phosphorylates Moesin in the context of border cell migration. However, I find some of the major claims not well supported. The authors claim that Msn is a master regulator. However, since Msn (and P-Moesin) is uniformly located along the periphery of the border cell cluster, it remains unclear whether Msn has an instructive role during border cell migration (distinguishing front from rear). Moreover, the authors claim that Moesin regulates contractile forces, but it seems equally possible based on the authors' data that Moesin is required for cortical stiffness (as shown previously in a different context, e.g. Kunda et al., 2008). Moreover, on the mechanistic level, it remains unclear how Msn activates Moesin.

As we recognized in our answer to reviewer #2, the use of "master regulator" was improper and we have now removed it from our manuscript. Line 24.

We do not argue though that Moesin regulates contractile forces, actually, we precisely argue the contrary: Msn regulates contractility independently of Moesin. However, we agree that Moesin probably regulate stiffness and changed most of the reference of “tension” into cortical stiffness. Finally, our data show that Msn i) can phosphorylate Moe *in vitro*, ii) that its kinase activity is required for BC migration and iii) that its depletion leads to a decrease of p-Moesin *in vivo*. As such, we think that the sum of these observations makes a clear case for Msn phosphorylating Moesin in BCs.

Specific comments

-Fig. 1. The authors claim that Msn is required for Moesin phosphorylation and border cell migration based on two RNAi lines. Since RNAi can have unspecific effects, the authors should use available *msn* alleles to corroborate their findings.

Working with the null alleles lead to other phenotypes including the abnormal differentiation of BCs [4]. RNAi allows us to avoid differentiation issues and to analyze *msn* loss-of-function of in normally differentiated clusters. Furthermore, our rescue experiments clearly show that the RNAi phenotype is specific as expression of Msn, but not of catalytically inactive Msn rescue migration.

-Fig. 1. The authors show that Msn phosphorylates the CERMAD domain of Moesin. Does Msn phosphorylate Thr556, which is required for unfolding and activation of Moesin?

We have now repeated our kinase assay with a CERMAD domain where Thr556 was mutated to an alanine. The phosphorylation of CERMAD is then abolished, demonstrating that Msn phosphorylate Thr556. This data is shown in Supplemental Fig. 2a and described at line 91.

-Fig. 1e. The authors claim that wild-type, but not a kinase-dead version of Msn rescues *msn*-RNAi flies. As a control, the authors should test whether Msn-K160D is expressed at a comparable level as the wild-type Msn protein. Moreover, it appears that Msn-K160D enhances the border cell migration defects of *msn*-RNAi flies. Do the authors think that Msn-K160D acts as a dominant-negative?

As the number of BCs expressing Msn is not sufficient compared to the rest of the tissue, we cannot assess this directly. However, expression of the different rescue in other tissues (and with another Gal4 line) shows that our transgenic line induces similar expression levels (Supplemental Fig. 2c).

Furthermore, we show that wild type, but not kinase dead Msn restore the phosphorylation of Moesin in Msn depleted clusters *in vivo* (Supplemental Fig. 2b, line 101-103)

It seems that expression of the KD indeed exacerbate the phenotype, but its expression alone does not induce a migration defect (7th column of Fig. 1e).

- Fig. 1. The authors claim that their data shows that Msn Phosphorylates Moesin to promote border cell migration. This conclusion would be much strengthened if the authors could show that an unphosphorylatable form of Moesin would fail to rescue moesin deficiency in border cells.

We have now performed this experiment and found that there is no rescue (Figure for Reviewer #3 a-b). We used a previously published UASp-MoeT556A line. The transgene is expressed through UASp that might express at lower levels in somatic tissues than the UAS promoter we usually use in BCs. The level of expression of MoeT556A, as assessed by a myc staining, seems close to the level of expression of the MoeT556D, but we preferred not to include these observations in the revised manuscript.

-Fig. 1. P-Moesin is not only present at the periphery of the border cell cluster, but also on the membranes of neighboring cells (nurse cells). In the authors' experiments, Msn is depleted in the border cells resulting in a reduction of P-Moesin in these cells. Does Msn specifically phosphorylate Moesin in the border cell cluster, or is this phosphorylation more widespread (incl. the nurse cell)?

Depletion of Msn is restricted to the border cells as the driver we use is not expressed in nurse cells. Accordingly, we see no obvious reduction in pMoe at nurse cell to nurse cell junctions. However, we see sometimes a reduction of pMoe at the nurse cells – border cell junctions, but: 1) we do not have the resolution to determine which cell(s) contributes to the remnant signal, 2) nurse cells are separated from each others when border cells migrate through them and thus the observed pMoe signal is reduced in half, and 3) the passage of the border cells in between nurse cells might modify the cortex of nurse cells and detach active moesin. Determining the relative contribution of these different possibilities seems difficult if not impossible.

-Fig. 4a. The authors state that “As moesin generates cortical tension in its active state [10]... My reading of reference 10 is that Kunda et al. 2008 (Ref. 10) in fact used AFM to measure cortical stiffness (Young's modulus), which in mitotic cells depends on moesin. Moreover, a phosphomimetic variant, T559D- Moesin induced a >2-fold increase in cortical rigidity. This increased cortical rigidity was independent of Myosin activity.

See next point.

-Fig. 4b-d. The analysis of membrane-curvature probably reflects the formation of protrusions. It unclear how the authors infer from this that Msn regulates cortical tension. Membrane curvature likely depends on tension, but also on cortical stiffness. The authors need to distinguish between tension and stiffness.

We thank the reviewer to pointing to the difference between stiffness and tension. Indeed, we misused the term “tension” and changed it to “stiffness” in the manuscript.

-Fig. 4e-g. The authors conclude that Msn regulates protrusion dynamics, in addition to overall cortical tension around the cluster. To this reviewer, it would be more plausible to say that Msn-dependent cortical tension at the cluster periphery restricts protrusion formation. What makes the authors conclude that Msn has two distinct functions here?

We think that Msn-dependent stiffness but not tension regulates protrusion formation. Indeed, increasing tension by expressing active Rok in a Msn depleted background does not restrict protrusion formation. However, we think that Msn regulates protrusion morphology and half-life through Myosin II mediated contractility.

-Fig. 5e. How reproducible is the symmetric distribution of pMLC2 in the wild-type and its absence in the msn depleted situation? The authors seem to show only a single border cell cluster.

The symmetric distribution of pMLC2 at the basis of front protrusions is observed in 76% of controls and almost never observed after depletion of Msn. We added a figure panel showing the quantification (Fig. 5f). However as the morphology of each protrusion is different, we cannot pool data and we have to show individual line scan.

-Fig. 5. Panel 'f' seems to be missing.

We corrected this in the new version of the manuscript.

-Fig. S2. The authors claim that Msn localizes to cortical actin-rich structures including at the base and the tip of protrusions. However, there are many Life-Act spots that do not co-localize with Msn. Moreover, the authors speculate that Msn may regulate Myosin II activity at these spots. Is there any evidence for this?

We clearly see Msn localizing to some actin-rich structures, but we do not claim that it colocalizes to all the actin structures observed. We do not see the fact that some Lifeact structures are negative for Msn as a contradiction. However, we agree that we have no direct evidence that Msn regulate the activity of Myosin II. It does regulate its distribution but not its activity. We have rephrased this sentence.

-Fig. 6b. The authors claim that extension and contraction events are coordinated across the cluster. However, if the border cell cluster almost constantly retracts back (as the authors state in the text (line 216)), I do not see how these two events are coordinated.

Sorry that we were not clear, but we did not mean that the back constantly retracts and we have now modified that sentence: "These MDMS reveal that the front of control clusters protrude with brief moments of contraction, while retraction events occurs at the back, suggesting that extension and contraction events are coordinated across the cluster (Fig. 6b-e)." Line 218-220.

We see that the fastest events of contractility (retractions) are clearly temporally coordinated with the fastest events of extensions (see Fig. 6c, 60 minutes for example) and hence we think that our statement that these events are coordinated is correct. Note that this coordination is lost after Msn depletion, since then, fast extension speeds are met with no retraction (at 60 minutes, for example).

-Fig. 6c. This panel is confusing. In Fig.6b, the authors show that both the front and the back of a border cell cluster can extend and retract. Fig. 6c now shows velocities of extension and retraction. Do these velocities relate to the front/back end of the cluster?

No, Fig.6c display velocities for all the extension and contraction activities recorded for each time point around the entire cluster. We have now clarified this point in the figure and in the figure legend. Fig. 6c and Line 224.

1. Li, Q., et al., *Ingestion of Food Particles Regulates the Mechanosensing Misshapen-Yorkie Pathway in Drosophila Intestinal Growth*. Dev Cell, 2018. **45**(4): p. 433-449 e6.
2. Ramel, D., et al., *Rab11 regulates cell-cell communication during collective cell movements*. Nat Cell Biol, 2013. **15**(3): p. 317-24.
3. Houalla, T., et al., *The Ste20-like kinase misshapen functions together with Bicaudal-D and dynein in driving nuclear migration in the developing drosophila eye*. Mech Dev, 2005. **122**(1): p. 97-108.
4. Cobreros-Reguera, L., et al., *The Ste20 kinase misshapen is essential for the invasive behaviour of ovarian epithelial cells in Drosophila*. EMBO Rep, 2010. **11**(12): p. 943-9.

Figure for Reviewer #1

a

a

b

REVIEWERS' COMMENTS:

Reviewer #1 (Remarks to the Author):

Nature Communication NCOMMS-18-16985-T

"Misshapen coordinates protrusion restriction and actomyosin contractility during collective cell migration" by Plutoni et al.

The manuscript has improved and my points, as well as most points of the other two reviewers seem to have been addressed. I think that the research is very interesting and the manuscript overall excellent.

I have three comments though and a number of editorial points:

1) line 260-262: the authors say that they did not observe defects when expressing Rok-cat in a control background, however, Fig. 7g and h seem to suggest otherwise.

2) line 346: The authors state that they think "it is unlikely that MSN acts upstream of Rok or MLC2". However, this seems to be what the model of Figure 8b seems to suggest.

3) Sup Fig. 1: The msn RNAi lines were tested by the authors, however, it is not clear from the manuscript how effective any of the other RNAi constructs are. It would be helpful if references would be included in the table for those RNAi lines that have been positively tested before by other groups.

Minor errors:

- Figure 1: check carefully all panel letters/figure legends/corresponding text: there are some mix-ups with the panel letters.

- Fig. 6: move figure legend text (lines 469-473 to description of panel b. Also, Line 476 "shown in d" - should it not be 'b' ?

- Figure 8 seems not be cited in the text.

- Supp Fig 4 and 5 have the same title

- Fig 2: add (MSD) after mean square displacement

- please, check carefully all sample sizes. There seem to be several mistakes, such as line 414: $n < 47$ ($X < n < 47$); line 559: "n=20 and 27" but there should be 4 sample sizes; line 562 states $n > 50$ but Sup Fig 5c states $n < 50$; line 545 says $10 > n > 27$ -> change to $10 < n < 27$.

- several tools have not been referenced or mentioned in M&M, such as eGFP, lifeact-ruby, HS-Gal4. Also, how long was the HS and how long after the heatshock was the tissue dissected (see line 531)?

line 271: revise title, e.g.: Msn regulates various processes during cell migration ...

line 284-288: discussion 2nd paragraph is much better but still needs some editing. Also, the genetic term "complementation experiments" has a different meaning and should be replaced. What the authors analyze here are 'epistatic relationships'.

line 62 and line 272: change to: ..was previously shown to be involved

line 317: change to: When we analyzed the effect of Msn depletion

line 326: change to: ...Moesin was able to rescue the effect of dominant

- line 99; WB ?

- line 512: change "tension" to stiffness?

- line 227: "These observations confirm..." -> These observations are consistent with

- The text and figures 4a and 6a contain a considerable number of grammar mistakes regarding the use of plural versus singular (in particular but not exclusively in the highlighted yellow text). Also, "where" is used in some cases where it should be 'were' or 'was'.

- separation of numbers and units is missing in several places

- M&M screen: move experimental details from 'Drosophila genetics' (lines 593-....) to 'Ste-like kinase screen' (lines 623 to ...)

- lines 611 and 616: change to .. pUAS-attB

- line 631: add specifications of objective

- lines 673-674. I did not understand the sentence.

Reviewer #2 (Remarks to the Author):

The authors have responded thoroughly to the reviewer comments. The only remaining issue is that the model shown in Fig 8 is overly complex, limiting its utility. If the authors could distill this information into a simpler diagram, it would be more useful. Secondly no data in the paper directly address myosin II function, so it does not seem appropriate to include MyoII in the summary models and diagrams.

Reviewer #3 (Remarks to the Author):

The authors have satisfactorily addressed my comments.

REVIEWERS' COMMENTS:

Reviewer #1 (Remarks to the Author):

Nature Communication NCOMMS-18-16985-T

"Misshapen coordinates protrusion restriction and actomyosin contractility during collective cell migration" by Plutoni et al.

The manuscript has improved and my points, as well as most points of the other two reviewers seem to have been addressed. I think that the research is very interesting and the manuscript overall excellent.

I have three comments though and a number of editorial points:

1) line 260-262: the authors say that they did not observe defects when expressing Rok-cat in a control background, however, Fig. 7g and h seem to suggest otherwise.

The Fig. 7g and h shows only the Rok-cat expression in *msn* RNAi background, but not in a control background. The only figure that show Rok-cat expression in a control background is the Supplementary Figure 5, where no defects were observed as mentioned at line 260-262.

2) line 346: The authors state that they think "it is unlikely that MSN acts upstream of Rok or MLC2". However, this seems to be what the model of Figure 8b seems to suggest.

Indeed, our statement was incorrect. We wanted to emphasize that it is unlikely that Msn acts as a direct activator of Rok or Sqh since Msn does not impact pSqh intensity. It is possible that Msn regulates only their localization, or acts as a local activator of Rok/Sqh.

We have changed the sentence to : "While we found that Msn is involved in the spatial organization of Myosin II activity, we think that it is unlikely that it is involved in its direct activation, as Msn depletion does not affect total pSqh levels." We also removed the next sentence that is now partially redundant ("Msn could regulate the localization of active Myosin II").

3) Sup Fig. 1: The *msn* RNAi lines were tested by the authors, however, it is not clear from the manuscript how effective any of the other RNAi constructs are. It would be helpful if references would be included in the table for those RNAi lines that have been positively tested before by other groups.

We have added the references as requested (see supplemental references).

Minor errors:

- Figure 1: check carefully all panel letters/figure legends/corresponding text: there are some mix-ups with the panel letters.

Corrections have been made in main text.

- Fig. 6: move figure legend text (lines 469-473 to description of panel b. Also, Line 476 "shown in d" - should it not be 'b' ?

For more clarity we have modified the legend as:

a. Schematic representation explaining how the morphodynamic analysis of migrating clusters was performed using the ADAPT plugin of ImageJ. The plugin measures the displacement of the overall cluster according to its geometric center (orange dotted circle) and represents these quantifications as a color-coded kymograph. On this scheme, negative velocities / retraction events are represented by blue arrows whereas positive velocity / extensions events by red arrows. **b.** Morphodynamic map displays as a kymograph the segmental velocities calculated and color-coded for visualization, for each position of the periphery of a cluster (red colors, extension; blue colors, contraction). The center of the MDM depicts the dynamics at the front of the cluster (purple) while the side (light blue) and the back (green) of the cluster is split at both the top and bottom extremities of the MDM.

- Figure 8 seems not be cited in the text.

This has been corrected in the main text at line 279.

- Supp Fig 4 and 5 have the same title

This has been corrected.

-

Fig 2: add (MSD) after mean square displacement

It has been added.

- please, check carefully all sample sizes. There seem to be several mistakes, such as line 414: $n < 47$ ($X < n < 47$); line 559: "n=20 and 27" but there should be 4 sample sizes; line 562 states $n > 50$ but Sup Fig 5c states $n < 50$; line 545 says $10 > n > 27$ -> change to $10 < n < 27$.

This have corrected the sample sizes descriptions.

- several tools have not been referenced or mentioned in M&M, such as eGFP, lifeact-ruby, HS-Gal4. Also, how long was the HS and how long after the heatshock was the tissue dissected (see line 531)?

line 271: revise title, e.g.: Msn regulates various processes during cell migration ...

Titles in the Discussion has been removed to fit with editorial requests.

line 284-288: discussion 2nd paragraph is much better but still needs some editing. Also, the genetic term "complementation experiments" has a different meaning and should be replaced. What the authors analyze here are 'epistatic relationships'.

We simplified the discussion of this paragraph and removed the reference to "complementation experiments".

line 62 and line 272: change to: ..was previously shown to be involved

It has been corrected as suggested by the reviewer.

line 317: change to: When we analyzed the effect of Msn depletion

It has been corrected as suggested by the reviewer.

line 326: change to: ...Moesin was able to rescue the effect of dominant

It has been corrected as suggested by the reviewer.

- line 99; WB ?

We replaced WB by "Western Blot".

- line 512: change "tension" to stiffness?

"Tension" has been replaced by "stiffness".

- line 227: "These observations confirm..." -> These observations are consistent with

It has been corrected as suggested by the reviewer.

- The text and figures 4a and 6a contain a considerable number of grammar mistakes regarding the use of plural versus singular (in particular but not exclusively in the highlighted yellow text). Also, "where" is used in some cases where it should be 'were' or 'was'.

- separation of numbers and units is missing in several places

- M&M screen: move experimental details from 'Drosophila genetics' (lines 593-...) to 'Ste-like kinase screen' (lines 623 to ...)

- lines 611 and 616: change to . pUAS-attB

- line 631: add specifications of objective

- lines 673-674. I did not understand the sentence.

All these points have been corrected as suggested by the reviewer.

Reviewer #2 (Remarks to the Author):

The authors have responded thoroughly to the reviewer comments. The only remaining issue is that the model shown in Fig 8 is overly complex, limiting its utility. If the authors could distill this information into a simpler diagram, it would be more useful. Secondly no data in the paper directly address myosin II function, so it does not seem appropriate to include MyoII in the summary models and diagrams.

We have slightly simplified the diagram and we have replaced “Myosin II” by “Spatial regulation of Myosin II”.

Reviewer #3 (Remarks to the Author):

The authors have satisfactorily addressed my comments.